# Development of a CMAQ-PMF-based composite index for prescribing an effective ozone abatement strategy: A case study of sensitivity of surface ozone to precursor VOC species in southern Taiwan

Jackson Hian-Wui Chang[1,2], Stephen M. Griffith[1#], Steven Soon-Kai Kong[1], Ming-Tung Chuang[3], Neng-Huei Lin[1,4,*]

[1]Department of Atmospheric Sciences, National Central University, Taoyuan, 32001, Taiwan
[2]Preparatory Center for Science and Technology, University Malaysia Sabah, Jalan UMS, 88400, Kota Kinabalu, Sabah, Malaysia
[3]Research Center for Environmental Changes, Academia Sinica, Taipei 11529, Taiwan
[4]Center for Environmental Monitoring and Technology, National Central University, Taoyuan, 32001, Taiwan
[#]Presently affiliated with Department of Atmospheric Sciences, National Taiwan University, Taipei 10617, Taiwan

*Correspondence to*: Neng-Huei Lin (nhlin@cc.ncu.edu.tw)

**Abstract.** Photochemical ozone pollution is a serious air quality problem under weak synoptic conditions in many areas worldwide. Volatile organic compounds (VOCs) are largely responsible for ozone production in urban areas where $NO_x$ mixing ratios are high while usually not a limiting precursor to ozone. In this study, the Community Multiscale Air Quality model – Higher-order Direct Decoupled Method (CMAQ-HDDM) at an urban-scale resolution (1.0 km x 1.0 km) in conjunction with positive matrix factorization (PMF) was used to identify the dominant sources of highly sensitive VOC species to ozone formation in southern Taiwan, a complex region of coastal urban and industrial parks and inland mountainous areas. First-order, second-order and cross sensitivities of ozone concentrations to domain-wide (i.e. urban, suburban and rural) $NO_x$ and VOC emissions were determined for the study area. Negative (positive) first-order sensitivities to $NO_x$ emissions are dominant over urban (inland) areas, confirming ozone production sensitivity favors the VOC-limited regime ($NO_x$-limited regime) in southern Taiwan. Furthermore, most of the urban areas also exhibited negative second-order sensitivity to $NO_x$ emissions, indicating a negative $O_3$ convex response where the linear increase of $O_3$ from decreasing $NO_x$ emissions was largely attenuated by the non-linear effects. Due to the solidly VOC-limited regime and the relative insensitivity of $O_3$ production to increases or decreases of $NO_x$ emissions, this study pursued the VOC species that contributed the most to ozone formation. PMF analysis driven by VOCs resolved 8 factors including mixed industry (21%),

vehicle emissions (22%), solvent usage (17%), biogenic (12%), plastic industry (10%), aged air mass (7%), motorcycle

exhausts (7%), and manufacturing industry (5%). Furthermore, a composite index that quantitatively combined the CMAQ-

HDDM sensitivity coefficient and PMF resolved factor contribution, was developed to identify the key VOC species that

should be targeted for effective ozone abatement. Our results indicate that VOC control measures should target on (1)

solvent usage for painting, coating and the printing industry, which emits abundant toluene and xylene, (2) gasoline fuel

vehicle emissions of n-butane, isopentane, isobutane and n-pentane, and (3) ethylene and propylene emissions from the

petrochemical industry.

## 1. Introduction

Photochemical production of tropospheric ozone ($O_3$) depends in a nonlinear manner on the availability of nitrogen oxides

($NO_x$) and volatile organic compounds (VOCs). Understanding ozone formation sensitivity to $NO_x$ and VOC emissions is

key to developing effective abatement strategies on $O_3$ pollution in heavily-polluted cities. Brute-force method (BFM) has

often been used to address the relationship between $O_3$ and its precursors (Hakami et al., 2004; Li et al., 2013; Zhang et al.,

2009). In a BFM approach within a 3D-modeling framework (e.g. Community Multiscale Air Quality modeling system –

CMAQ), individual emissions are perturbed by a small amount and the model response is recorded against the baseline,

representing the sensitivity coefficient. However, the linear response of BFM is insufficient to account for changes in

secondary pollutants (i.e. ozone) generated by nonlinear interactions of various substances. Therefore, more sophisticated

numerical techniques have been introduced such as the adjoint method (Hakami et al., 2006; Wang et al., 2021), Decoupled

Direct Method in Three Dimensions – DDM 3D (Dunker et al., 2002b; Luecken et al., 2018), and higher-order decoupled

direct method – HDDM 3D (Cohan et al., 2005; Koplitz et al., 2021), where the 3D aspect is specific to implementation in

chemical transport models (CTMs) with a 3D-modeling framework (e.g. CMAQ). These methods offer an alternative to

BFM by directly solving the auxiliary equations that represent the change in concentration over change in emission (dC/dE)

derived from the governing equations of the model.

DDM 3D, using a semi-implicit finite difference scheme with different time steps (Hakami et al., 2004; Yang et al., 1997),

has been widely implemented in CTMs (Dunker et al., 2002b, 2002a) to calculate the first-order sensitivities (i.e. tangential,

$\partial C/\partial E$) of ozone with respect to initial concentration, boundary concentrations, and precursor emission rates. First-order

sensitivity describes the linear response of the model to a perturbed input parameter, while higher-order sensitivity describes

the nonlinear quadratic, cubic and higher-power responses. In the presence of strong nonlinearity (e.g. in transition between

VOC-limited and NOx-limited regime), first-order sensitivity alone may be insufficient to characterize the response if the

magnitude of the emissions changes are large (i.e. a manifestation of the nonlinear chemistry involved in ozone formation)

and second-order sensitivity is necessary to provide the additional nonlinear responses of the system (Xing et al., 2011; Xiao

et al., 2010).  Thus, in HDDM, the sensitivity is calculated by solving the first-order ($\partial C/\partial E$) and second-order (i.e. local

slope, $\partial^2 C/\partial E^2$) derivatives in the model when a relative perturbation (e.g. ±10%) is applied to a targeted parameter while

keeping all other factors constant.

Apportioning pollutant concentrations to their sources is an ideal strategy to guide emission control policy (Dunker et al.,

2015; Liu et al., 2008). However, source apportionment of a secondary pollutant such as ozone is complex, where a

relatively crude method is to simply conduct simulations with and without $NO_x$ and VOC emissions from a given source so

that the difference in ozone concentration is a measure of the source contribution (Bergin et al., 2008). Other source

apportionment approaches such as ozone source apportionment technology (OSAT) and integrated source apportionment

method (ISAM) are embedded into air quality models (Comprehensive Air Quality Model with Extensions (CAMx) and

CMAQ, respectively) and rely on tagging $NO_x$ and VOCs as tracers from emission to ozone production to estimate the

contributions of different sources to the eventual ozone concentration (Dunker et al., 2002a; Kwok et al., 2015). Unless a

high resolution sector profile emission inventory is available, the application of OSAT and ISAM is often limited to four

main sector groups: on-road, non-point (area), point, and biogenic sources (Wang et al., 2009; Li et al., 2013a). In addition,

the spatio-temporal and sector group distribution uncertainties associated with the emission inventory greatly influence the

accuracy of these source apportionment methods (Zheng et al., 2009).

Positive matrix factorization (PMF), a receptor-based approach, offers an alternative to those embedded methods by

generating a set of ozone precursor source composition profiles, each identifying a mix of compounds associated with a

particular category of emissions. Driven by measured concentrations at receptor sites, PMF can be used as an ozone source apportionment method directly and is independent of the uncertainties associated with the emission inventory. In recent years, PMF has been widely used to estimate VOC apportionment because it requires only identifying characteristics of the source profiles to interpret the PMF factors (Ji et al., 2022; Fan et al., 2021; Huang and Hsieh, 2020; Chen et al., 2019; Wu et al., 2016). For instance, Pallavi et al. (2019) found that traffic contributed 47% of the total benzene in India followed by residential biofuel use and waste disposal (25%) and industrial emissions and solvent use (20%). Zhao et al. (2020) used PMF in conjunction with an ozone formation potential (OFP) calculation to infer that VOCs from industrial and vehicular emissions were the dominant ozone precursors in Nanjing, East China, particularly xylenes, toluene, and propene. Chen et al. (2019) used PMF to resolve the dominant VOC sources at an industrial complex in central Taiwan and found that the monitored VOC concentrations of vehicle exhaust, solvent use, and diesel attributed to high OFP were associated with easterly to southeasterly winds. Huang and Hsieh (2020) analyzed VOC data by PMF from western coastal Taiwan and suggested that on a mass concentration basis industrial emissions are the greatest contributors to OFP. However, previous PMF-OFP studies have only identified key VOC sources to OFP without investigating the sensitivity of OFP to these VOC emissions and thus failed to provide comprehensive advice on which sources to prioritize for effective ozone abatement.

In this work, we used CMAQ-HDDM-3D in conjunction with PMF, enabling us to interpret the sensitivity and source apportionment analyses together for a more comprehensive investigation of the key VOC sources to OFP. We combined the two approaches as a novel methodology to provide additional insights on the dominant sources of highly sensitive VOC species to ozone formation in a VOC-limited urban area of Taiwan, but should be widely applicable across urban areas that experience similar $O_3$ episodes such as Hong Kong (Ling and Guo, 2014), Beijing and Hebei (Chi et al., 2018), Mexico City (Lei et al., 2007), Houston, United States (Mazzuca et al., 2016). Specifically, HDDM describes the $O_3$ sensitivity response to a speciated emission and PMF identified the sources of these species for effective ozone abatement strategy. While most of the HDDM sensitivity studies are performed at coarse resolution (>4.0 km), this study was conducted at urban-scale resolution (1.0 km x 1.0 km). We investigated the first-order, second-order, and cross sensitivities of ozone concentrations to domain-wide $NO_x$ and VOC emissions and provide an overview of the $O_3$-precursors sensitivity across the study area. We

then identified the VOC species to which ozone formation was most sensitive. Finally, we mapped these highly sensitive VOC species to the PMF model source apportionment and identified their dominant sources for effective emission control strategies using a composite index of CMAQ sensitivities and PMF factor contributions. The results of our study provide important information as to which VOC species are key to ozone formation and where to reduce sources of these VOC species for effective ozone abatement.

## 2. Methods

### 2.1 Study Period & Area

In this study, we selected the period from 07-20 Oct. 2018 for conducting simulations of photochemical ozone production and transport in southern Taiwan. The selected case in October 2018 is the seasonal transition period when the summer season is in transition to the winter. The case can reasonably represent the typical ozone pollution conditions during seasonal transition period in Taiwan because the synoptic weather pattern of the event features a weak intrusion of Asian continental anticyclone system which slowly propagated eastward causing the prevailing wind in Taiwan dominated by weak northeasterly (NE) flows due to continental high-pressure peripheral circulation (see Figure S1). Hsu & Cheng (2019) identified six synoptic weather patterns common in Taiwan and studied the characteristics of corresponding air pollutants in each pattern using 6 years (2013-2018) daily averaged wind fields and sea-level pressure observed at surface weather stations in Taiwan. Among the six patterns (C1-C6), C4 has the highest mean $O_3$ concentrations and occurs predominantly in October. It features a weak anticyclone over the Asian continent and the Pacific subtropical high-pressure system does not have an apparent influence in Taiwan, which is similar to our selected case in October 2018. Although the photochemistry is strong in summer season, the seasonal $O_3$ variation in Taiwan shows that the monthly O3 concentration is relatively higher during the seasonal transition months (i.e., October) compared to other seasons (Hsu and Cheng, 2019; Chen et al., 2021; Cheng et al., 2015). This is because during the seasonal transition months, when the photochemical reaction is still strong compared to that of the winter months together with the reduced ventilation capability, the $O_3$ concentration can accumulate (Yen and Chen, 2000; Tsai et al., 2008) (See Figure S2 for ozone seasonality in Taiwan). During the seasonal transition period in autumn, southern Taiwan often suffers from high $O_3$ episodes (Hsu and Cheng, 2019; Chen et al., 2004, 2021).

Regional synoptic weather in autumn usually features a weak anticyclone over the Asian continent, allowing for local accumulation of pollutants, while the Pacific subtropical high-pressure system has shifted eastward with no apparent influence on Taiwan (Hsu and Cheng, 2019). This synoptic weather pattern is also categorized as High Pressure Pushing (HPP), which occurred when the leading edge of Asian continental high-pressure systems moves over China coastal provinces and carried pollutants southward towards downwind area in Taiwan (Chuang et al., 2008). In addition,

precipitation is less and vertical dispersion is weaker than in summer (Yen and Chen, 2000; Tsai et al., 2008), which contribute to relatively more frequent ozone episodes in autumn. These conditions are also common occurrences for high $O_3$ episodes in other places such as Hong Kong and East China cities in autumn (Lee et al., 2009; Liu et al., 2021; Yu et al., 2021). A 5-day spin-up period (02-06 Oct. 2018) was discarded from the analysis to eliminate the effects of initial conditions on $O_3$ simulations.


Since the 1990s, ozone concentrations have been increasing in many areas of southern Taiwan (Chang et al., 2005; Chen et al., 2021). Kaohsiung city, the biggest city in southern Taiwan and second biggest in all of Taiwan, hosts many heavy industries, including petrochemical plants, refineries, steel-making plants, and power generation plants. Xiaogang (XG) district is located in the southeastern portion of the city and hosts a petrochemical industrial park with an overall area of 403

hectares. Three power plants, each producing 1200-4300 MW day$^{-1}$, are located within a distance of 35 km of XG, and lead to some of the poorest air quality in Taiwan.

**2.2 WRF-CMAQ Model Configuration**

WRF model version 3.9.1 and CMAQ model version 5.2.1 (WRF-CMAQ) is used in the current study with a domain

configuration of a four-nested grid system centered at 28°N and 125°W (Figure 1a). The outermost domain (D01) covers most of mainland China with a horizontal resolution 27 km x 27 km and 166 x 169 grids, which is then nested to the second domain (D02) of 9 km x 9 km and 223 x 223 grids over East China. The third domain (D03) covers the whole island of Taiwan with a resolution of 3 km x 3 km and 223 x 223 grids and the innermost domain (D04) focuses on southern Taiwan

with a 1 km x 1 km resolution and 136 x 136 grids. There are 41 vertical sigma layers spaced unequally from the ground to the model top 50 hPa, with the bottom 8 layers resolved below 1.5 km (Figure S3).

The initial and boundary meteorological conditions are adopted from the National Centers for Environmental Prediction (NCEP) global final analysis (FNL) data at 0.25° x 0.25° resolution updated every 6 hours. The projected 2017-year Multi-resolution Emission Inventory for China (available at http://meicmodel.org) with 0.25° x 0.25° resolution is used for D01 and D02, while 2016 Taiwan Emission Data System (TEDS) version 10 (Figure 1(c, d)) is used for D03 and D04. Biogenic emissions are calculated offline using Model of Emissions of Gases and Aerosols from Nature (MEGAN) version 2.10 (Guenther et al., 2006).

Other than local anthropogenic emissions, the contribution of long-range transport (LRT) from East Asia (e.g. Chinese emissions) is also substantial to Taiwan's air quality especially under strong northeasterly winds condition (Wu and Huang, 2021; Chang et al., 2022a). Lin et al. (2004) identified three types of common LRT events in Taiwan: (1) dust storm (DS), LRT with pollutants (frontal pollution; FP), and (3) LRT of background airmasses (BG). When there is no frontal system, local pollution (LP) dominates the air quality in Taiwan. During wintertime and springtime, the occurrence of LP cases were 70% and about 30% were LRT cases (Lin et al., 2004). Lin et al. (2005) estimated that the long-range transport of pollutants contributes to about 30 μg m$^{-3}$, 230 ppb and 0.5 ppb to the $PM_{10}$, CO, and $SO_2$ concentrations, respectively, in northern and eastern Taiwan. Meanwhile a smaller contribution is estimated in southern Taiwan due to the geographic  (Lin et al., 2005).

In our work, the oceanic chlorine emission is calculated online by CMAQ as a function of meteorology using an OCEAN file which specifies the fraction of each grid cell that is open ocean (OPEN) or surf zone (SURF). Figure 2a-c presents the spatial distribution of CMAQ calculated sea-salt aerosol cations (ASEACAT - $Na^+$, $K^+$, $Ca^{2+}$ and $Mg^{2+}$), fine-mode chlorine and coarse-mode chlorine emission rates averaged during the entire simulation period. The sea-spray emissions were higher in the surf zone area and highest emission rates were found along the eastern offshore of southern Taiwan. This is because of the enhanced formation of sea-spray aerosols associated with higher relative humidity and greater offshore winds along the

eastern offshore of southern Taiwan that is open to the Western Pacific Ocean. Besides, the anthropogenic chlorine emissions (PCL) are obtained from TEDS v10 emissions, and they are concentrated over the heavily industrialized urban areas of southern Taiwan (see Figure 2d).

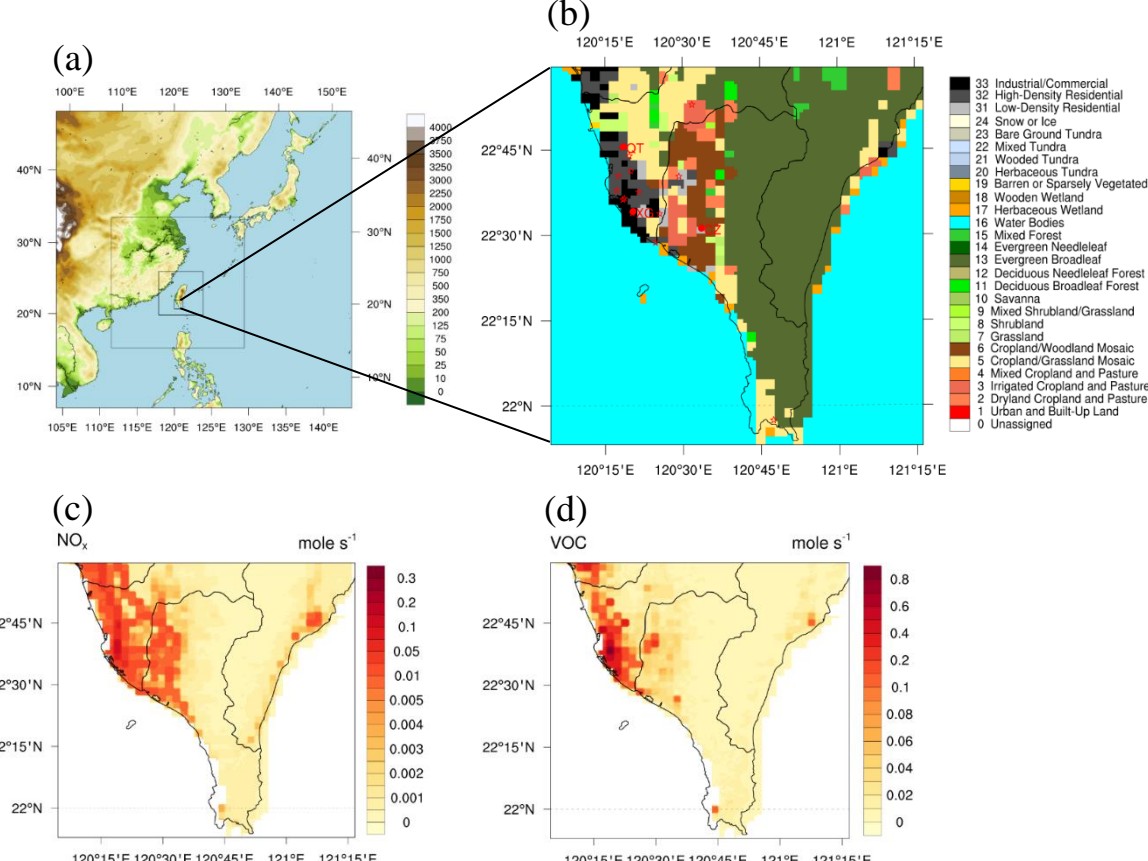

Figure 1: (a) Domain configuration of four-nested grid system, (b) land use of the innermost domain; "urban" and "inland" areas are represented by Class 31, 32, 33 and Class 6, respectively, (c,d) monthly averaged $NO_x$ and VOC emissions in the innermost domain obtained from 2016 TEDS-10 emission inventory. The location of each TEPA air quality stations (red stars) and PAMS stations (red dots with label) used in the current study are displayed in the innermost domain. Refer Figure S3 and Table S3 for details of each TEPA and PAMS station.


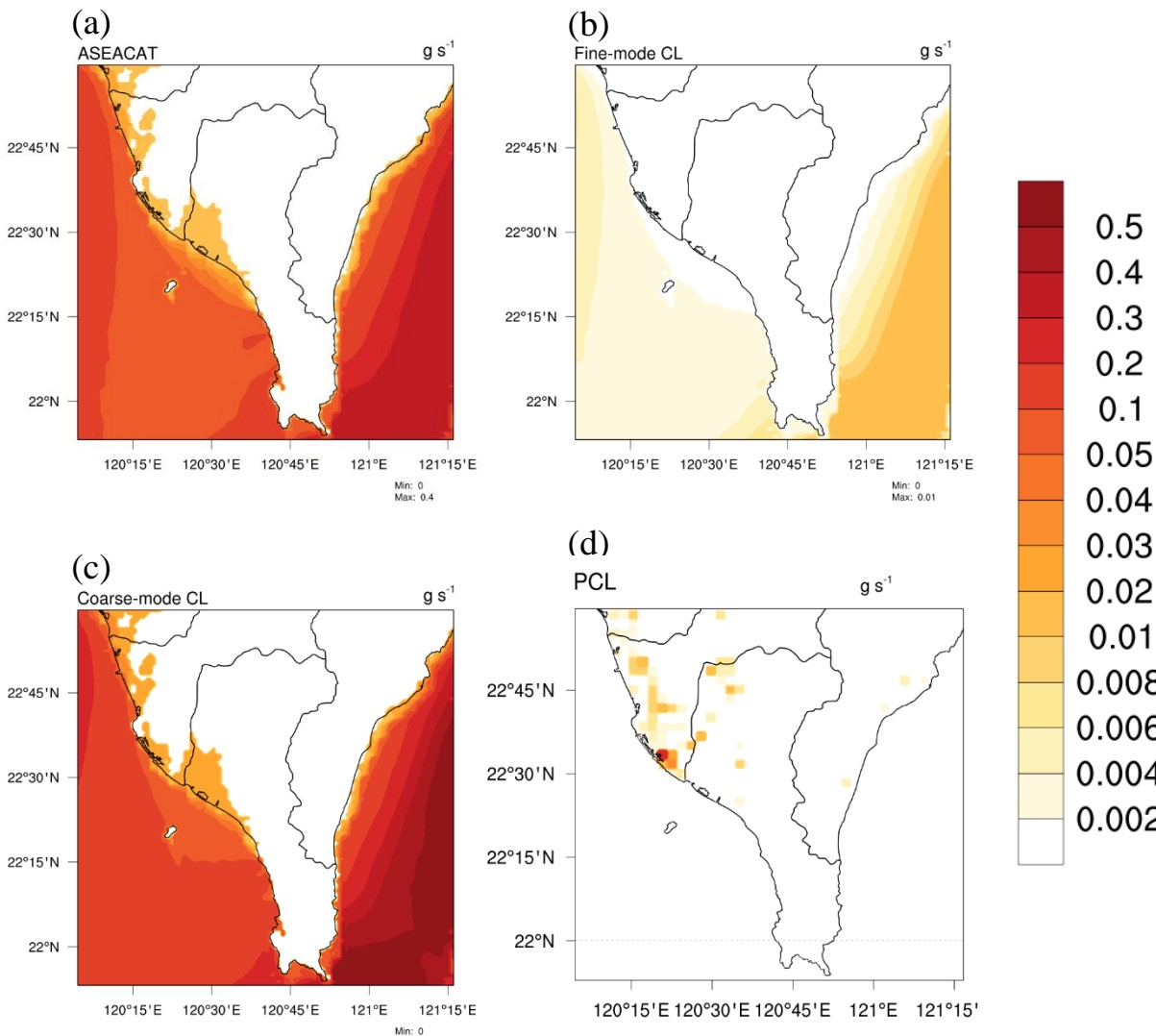

*Figure 2: CMAQ calculated (a) sea-salt aerosol cations (ASEACAT) emissions (Na⁺, K⁺, Ca²⁺ and Mg²⁺), (b) fine-mode chlorine SSA emissions, (c) coarse-mode chlorine SSA emissions and (d) TEDS v10 anthropogenic chlorine (PCL) emissions averaged during the entire simulation period.*


*To obtain more accurate dynamical downscaling, grid nudging was applied in the coarse domain D01 and D02; observation nudging was applied in the fine domain D03 and D04. Grid nudging is applied to the horizontal wind components, potential temperature, and water vapor mixing ratio; it is only applied above the PBL. The observational data for observation*

*nudging include hourly surface observations such as atmospheric pressure, air temperature, relative humidity, wind speed and wind direction from 36 surface meteorological stations (https://www.epa.gov.tw/), and the twice-daily at 00:00 and 12:00 UTC sounding data such as potential height, temperature, dew point temperature, RH, wind direction, wind speed at each specified isobaric level from 2 radiosonde observation stations in Taiwan. The nudging coefficients, which determine the strength of the assimilation tendency term were set to be $6 \times 10^{-4}$ for observation nudging and $3 \times 10^{-4}$ for grid nudging.*

*These values of coefficients were recommended by the WRF user guide and tested to be appropriate in previous studies (Li et al., 2022; Borge et al., 2008).*

*The simulation adopted the Carbon-Bond Mechanism CB6 (Yarwood et al., 2010), which was developed as an update to CB05 to provide a condensed chemical mechanism for use in atmospheric models. CB6 includes five additional organic*

*compounds that are long-lived and relatively abundant (i.e. propane, acetone, benzene, ethyne, and higher ketones) and a more detailed representation of organic nitrate reactions. CB6 has been tested against measurements over a wide variety of spatial, temporal, chemical and meteorological conditions and has shown good agreement with measurements of ozone and nitrogen oxides in both urban and rural areas (Luecken et al., 2019). The halogen chemistry in CB6 considers chlorine-related reactions such as $ClNO_2$, $HCl$ and $HNO_3$ production from heterogeneous uptake of $N_2O_5$ on the aerosol surface,*

*which are important to ozone pollution over coastal cities in southern Taiwan.*

*The Noah land surface model (LSM) is used to describe the land-atmosphere interaction (Chen and Dudhia, 2001). The urban effect in Noah LSM is invoked by implementing a single-layer urban canopy model (UCM), which assumes infinitely long 2D street canyon urban geometry to improve the processes associated with the exchange of momentum, heat, and*

*moisture in the urban environment (Kusaka and Kimura, 2004; Kusaka et al., 2001). To take full advantage of the UCM*

scheme, three additional urban classes (i.e. low-density residential, high-density residential and industrial/commercial) are further classified for better representation of the urban features (Figure 1b).

The asymmetric convective model version 2 (ACM2) boundary layer scheme is selected to represent the boundary layer
process (Pleim, 2007). It includes the nonlocal scheme of the original ACM combined with an eddy diffusion scheme. Thus, ACM2 is able to represent both the supergrid- and subgrid-scale components of turbulent transport in the convective boundary layer. Also used were the Goddard Cumulus Ensemble (GCE) microphysics scheme (Tao et al., 2003), Radiative Transfer Model (RRTM) longwave radiation scheme (Gallus and Bresch, 2006), Goddard shortwave radiation scheme (Chou and Suarez, 1999), Monin-Obukhov similarity scheme, and Kain-Fritsch cumulus parameterization scheme (only at D01 and
D02).

The modeled results of both meteorology (i.e. 2m-temperature, wind speed & direction and relative humidity) and air quality (i.e. ozone, nitrogen oxides and volatile organic compounds) are validated against 15 Taiwan Environmental Protection Administration (TEPA) air quality stations. Overall, the modeling system reproduced the observed meteorological and air
quality conditions within the benchmark values (see Supplementary Material – Model Evaluations).

The "urban" and "inland" grid cells are defined according to the USGS-24 Land Use Category. "Urban" area is represented by Class 1 - Urban and Built-up Land, which we further classified into Class 31, 32, 33 (see Figure 1b) for WRF Single-Layer Urban Canopy Scheme (SLUCM) simulation. We refer the readers to our previous work for detailed discussion on the
land use classification and SLUCM implementation in (Chang et al., 2022b). "Inland" area is represented by Class 6 – Cropland/Woodland Mosaic (see Figure 1b). The general meteorological pattern of the event features a weak intrusion of Asian continental anticyclone system which slowly propagated eastward causing the prevailing wind at synoptic scale in Taiwan dominated by weak northeasterly (NE) flows due to continental high-pressure peripheral circulation (see Figure S1). At local scale in southern Taiwan, the steering of weak NE flows by the orographic effect of the Central Mountain Range

(CMR) enhanced the local circulations (i.e. land-sea breeze), and eventually pushed the locally produced urban O$_3$ as well as

its precursors NO$_x$ and NMHC towards the inland areas (see Figure 3).

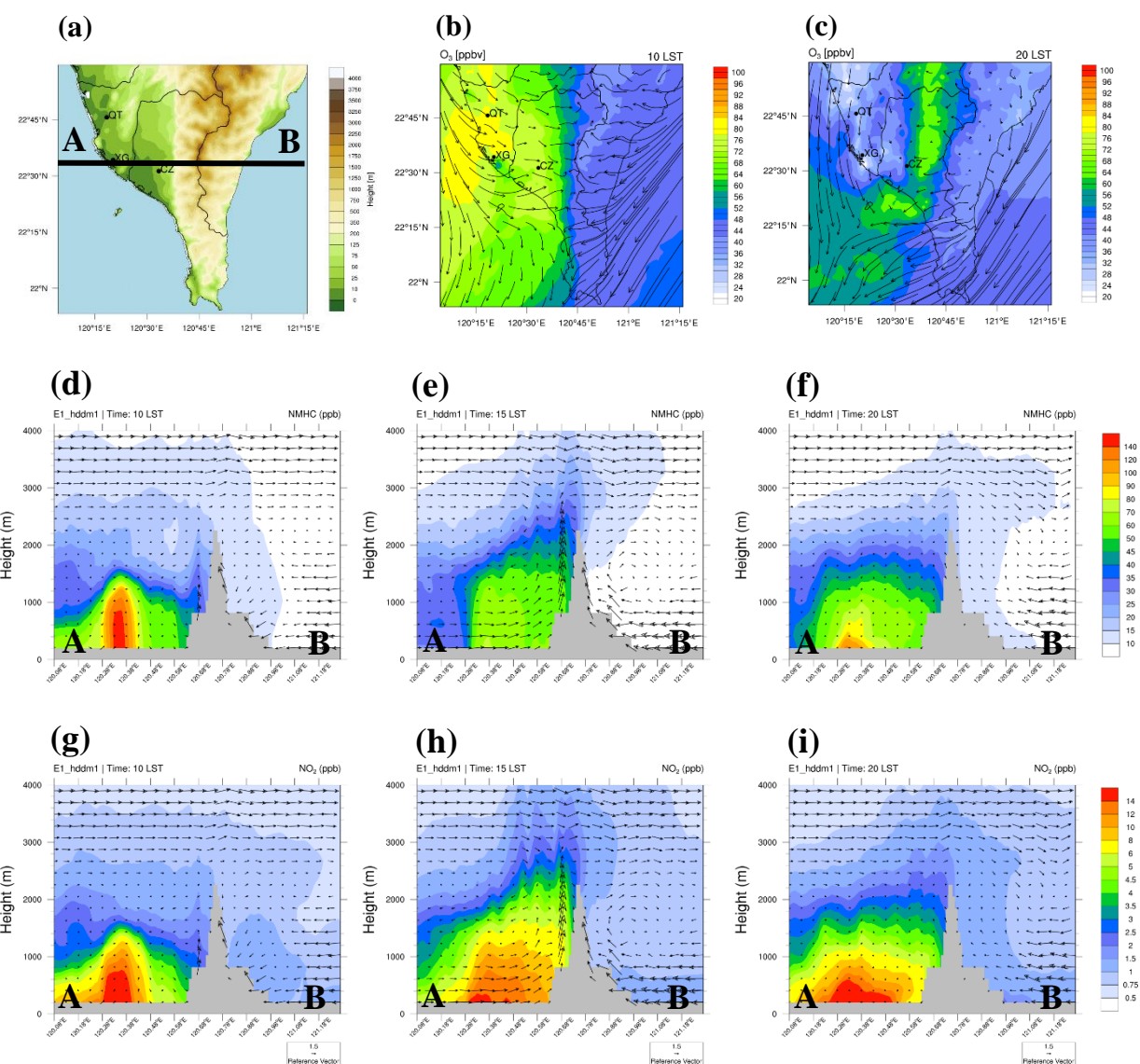

Figure 3: (a) Topography of the innermost domain. (b, c) Spatial distribution of O$_3$ concentration averaged during the entire

simulation at 10 LST and 20 LST, respectively. (d-f) Vertical profile of NMHC concentration cross sectioned at AB (see

Figure 3a) averaged during the entire simulation at 10 LST, 15 LST, and 20 LST, respectively. (h-i) Same as d-f but for $NO_2$ concentration.

## 2.3 Higher-order Decoupled Direct Method (HDDM)

The response of a chemical concentration to perturbations in model parameters (i.e. emissions, initial condition, boundary condition and reaction rate constants) can be evaluated through sensitivity analysis. A perturbed sensitivity parameter, $p_i$, is related to the unperturbed sensitivity parameter, $P_i$, in the baseline simulation as:

$$p_i = \varepsilon_i P_i = (1 + \Delta \varepsilon_i) P_i \tag{1}$$

where $\varepsilon_i$ is a scaling factor with a nominal value of 1.0 when there is no perturbation, and $\Delta \varepsilon_i$ is a perturbed scaling factor.

The response of a chemical concentration, $C$, against the perturbations in a sensitivity parameter, $p_i$, is defined as the sensitivity coefficient, $S_i$. The first- and second-order sensitivity coefficients, $S_i(1)$, and $S_{i,j}(2)$ are defined as follows:

$$S_i^{(1)} = P_i \frac{\partial C}{\partial p_i} = P_i \frac{\partial C}{\partial (\varepsilon_i P_i)} = \frac{\partial C}{\partial \varepsilon_i} \tag{2}$$

$$S_{i,j}^{(2)} = P_i \frac{\partial C}{\partial p_i} P_j \frac{\partial C}{\partial p_j} = P_i \frac{\partial C}{\partial (\varepsilon_i P_i)} P_j \frac{\partial C}{\partial (\varepsilon_j P_j)} = \frac{\partial^2 C}{\partial \varepsilon_i \partial \varepsilon_j} \tag{3}$$

Both $S_i(1)$, and $S_{i,j}(2)$ have the same units as the chemical concentration, C. $S_i(1)$ measures the impact of one variable $p_i$ on a

concentration, C; $S_{i,j}(2)$ measures the impact of another variable $p_j$ on a first-order sensitivity, $S_i(1)$, which can be used to investigate the second-order cross sensitivity of the system. When $i=j$, $S_{i,i}(2)$ represents the local curvature of the relationship between a concentration and a single parameter, thus indicating the responsiveness of $C$ to a broad range of $p_i$ and describing a nonlinearity of the system. A second-order sensitivity of zero indicates a perfectly linear response, while greater values represent proportionally greater non-linearity effects. Matching signs of the sensitivities (e.g. positive first-order and positive

second-order) represent a convex response while opposite signs represent a concave response (Figure S4). In the context of our study, reduced emission is the focus so a convex response represents a non-linear ozone concentration decrease with decreasing emissions and a concave response represents a non-linear ozone concentration increase with decreasing emissions.

### 2.3.1 Taylor Series Expansion

After determining the sensitivity coefficients, we can apply these coefficients for estimating the concentration changes in emission reduction scenarios based on Taylor series expansion. The concentration change from any fractional perturbations in a sensitivity parameter are approximated by (Cohan et al., 2005):

$$C_i \mid_{p_i = P_i + \Delta \varepsilon_i P_i} \approx C_o \mid_{p_i = P_i} + \Delta \varepsilon_i S_i^{(1)} + \frac{1}{2} \Delta \varepsilon_i^2 S_{i,i}^{(2)} + \ldots + \frac{1}{n} \Delta \varepsilon_i^n S^{(n)} \tag{4}$$

where $C_i$ is the concentration when $p_i$ has been perturbed by an amount $\Delta \varepsilon_i P_i$. Subscript $i$ represents the targeted emission species (i.e. a $NO_x$ or VOC species). When multiple sensitivity parameters are perturbed simultaneously, a second-order Taylor approximation includes the interaction between the two parameters (Cohan et al., 2005):

$$C_{i+j} \cong C \mid_{p_{i,j} = P_{i,j} + \Delta \varepsilon_{i,j} P_{i,j}} \approx C_o + \Delta \varepsilon_i S_i^{(1)} + \Delta \varepsilon_j S_j^{(1)} + \frac{1}{2} \Delta \varepsilon_i^2 S_{i,i}^{(2)} + \frac{1}{2} \Delta \varepsilon_j^2 S_{j,j}^{(2)} + \Delta \varepsilon_i \Delta \varepsilon_j S_{i,j}^{(2)} + \ldots \tag{5}$$

where $S_{i,j}$ is the cross sensitivity between the two parameters, which differs from the sum of the two sensitivities, $S_i$ and $S_j$.

In this study, CMAQ-HDDM-3D *v5.2.1* is used to calculate the sensitivity coefficients to predict the $O_3$ response to perturbations in $NO_x$ and VOC emissions. Perturbations at ±10% were made on the domain-wide emissions of $NO_x$ and VOCs (D01-D04) from both anthropogenic and biogenic sources, but only sensitivity coefficients from the innermost domain (D04) are used for data analysis. A total of 25 sensitivity tests were performed to quantify the change in $O_3$ concentration due to the perturbations made in each test (Table 1); the first 5 sensitivity tests, S1-S5, account for the first-order, second-order and cross sensitivity due to the perturbed $NO_x$ and total VOC emissions and the other 20 sensitivity tests, S6-S25, account for sensitivities due to perturbed individual VOC model species. Noted that HDDM approach was only used in experiment S3-S5 which involves calculation of higher order sensitivity, while DDM approach was used in all other experiments and in conjunction with PMF analysis.

Table 1: Perturbed emissions considered in the 25 sensitivity tests. The first 5 sensitivity tests S1-S5 accounts for the first-order, second-order and cross-order sensitivity due to the domain-wide $NO_x$ and VOC emissions and the other 20 sensitivity tests S6-S25 accounts for the individual VOC model species.

| Exp | Perturbations | First-order sensitivity | Second-order sensitivity |
|---|---|---|---|
| S1 | $\partial NO_x$ | $NO_x$ emissions | - |
| S2 | $\partial VOC$ | VOC emissions | - |
| S3 | $\partial^2 NOx$ | - | $NO_x$ emissions |
| S4 | $\partial^2 VOC$ | - | VOC emissions |
| S5 | $\partial NO_x \partial VOC$ | - | $NO_x$, VOC emissions |
| S6 | $\partial ETHA$ | Ethane emissions | - |
| S7 | $\partial PRPA$ | Propane emissions | - |
| S8 | $\partial PAR$ | Paraffins emissions | - |
| S9 | $\partial ETH$ | Ethene emissions | - |
| S10 | $\partial ISOP$ | Isoprene emissions | - |
| S11 | $\partial TER$ | Monoterpene emissions | - |
| S12 | $\partial OLE$ | Terminal olefins emissions | - |
| S13 | $\partial IOL$ | Internal olefins emissions | - |
| S14 | $\partial FORM$ | Formaldehyde emissions | - |
| S15 | $\partial ALD2$ | Acetaldehyde emissions | - |
| S16 | $\partial ALDX$ | Higher acetaldehyde emissions | - |
| S17 | $\partial ACET$ | Acetone emissions | - |
| S18 | $\partial KET$ | Ketone emissions | - |
| S19 | $\partial ETHY$ | Acetylene emissions | - |
| S20 | $\partial ETO$ | Ethanol emissions | - |
| S21 | $\partial MEOH$ | Methanol emissions | - |
| S22 | $\partial BENZ$ | Benzene emissions | - |
| S23 | $\partial TOL$ | Toluene emissions | - |
| S24 | $\partial XYL$ | Xylene emissions | - |
| S25 | $\partial NAPH$ | Naphthalene emissions | - |

**2.4 Positive Matrix Factorization (PMF) Model**

Positive matrix factorization (PMF) model is a source-receptor statistical factor analysis method, widely used for apportioning sources of air pollution and resolves the dominant factor profile without prior knowledge of sources; the

measured data uncertainty is used to optimize the model (Daellenbach et al., 2017; Fountoukis et al., 2014; Fan et al., 2021). In this study, US EPA PMF 5.0 receptor model was used for the source apportionment of measured VOC species:

$$x_{ij} = \sum_{k=1}^{p} g_{ik} f_{kj} + e_{ij} \tag{6}$$

where $x_{ij}$ is the $j$-th species concentration measured in the $i$-th sample, $g_{ik}$ is the airborne mass contribution from the $k$-th source in the $i$-th sample, $f_{kj}$ is the $j$-th species fraction to the $k$-th source and $e_{ij}$ is the residual associated with the $j$-th species concentration measured in the $i$-th sample. $p$ is the total number of independent sources. Noted that the subscript "$i$" in Eq. (6) refers to the $i$-th sample which is different from Eq. (1-5) that refers to the emission of $i$-th species. In the source parsing

process, the objective function $Q$ is solved with an iterative minimization algorithm. The objective function is defined as:

$$Q = \sum_{i=1}^{n} \sum_{j=1}^{m} \frac{x_{ij} - \sum_{k=1}^{p} g_{ik} f_{kj}}{u_{ij}} \tag{7}$$

where $u_{ij}$ is the uncertainty of $j$-th species in $i$-th sample.

Figure 4 shows the overall framework of the PMF source apportionment methodology. The data used to drive the PMF

model source apportionment was obtained from Taiwan's Photochemical Assessment Monitoring Stations (PAMS); details of the PAMS location and sampling protocol are provided in the supplementary material. Species with >55% of samples missing data or below MDL were discarded (Chen et al., 2019; Wu et al., 2016) . Among 54 PAMS-VOC species, a total of 16 species were discarded due to abundant missing data, but had a low OFP (Figure S5). The sample data uncertainty is calculated using:

$$unc = \begin{cases} 5/6 \times MDL & , x_{ij} < MDL \\ \sqrt{(\sigma \times x_{ij})^2 + (0.5 \times MDL)^2} & , x_{ij} \geq MDL \end{cases} \tag{8}$$

where $MDL$ represents the minimum detection limit, $\sigma$ is the error fraction (i.e. 10% was used in this study), and $x_{ij}$ is the concentration of $j$-th species in $i$-th sample. In the PMF, we tested a range of 3-8 VOC sources to determine the optimal number. Details of this protocol will be discussed further below.

The Taiwan PAMS data provided a total of 54 VOC species, but not all species were used for the PMF model due to an abundance of values below MDLs. Among the selected 38 VOC species (n=744 samples), they were further categorized into three categories: strong, weak and poor according to two steps. The first step calculated the signal-to-noise ratio (S/N) where species with a S/N value less than 0.5 were classified as poor and removed from the PMF model. This threshold value 0.5 was determined according to the EPA PMF v5.0 user guide and also recommended by other PMF studies (Rajput et al.,

2016; Reff et al., 2007). Next, we performed the model base run and calculated the correlation between the modeled and measured concentration for each species. In this step, species with a correlation value ≥0.6 were classified as 'strong' and otherwise as 'weak', which were then down-weighted by tripling the analytical uncertainty (Pallavi et al., 2019; Chen et al., 2019). Details of the species categorization is summarized in Table 2. Finally, a total of 21 VOC species were identified and used in the PMF source apportionment analysis, which accounted for 75.0%, 76.4% and 76.1% of the total VOC

concentrations in CZ, QT and XG, respectively.

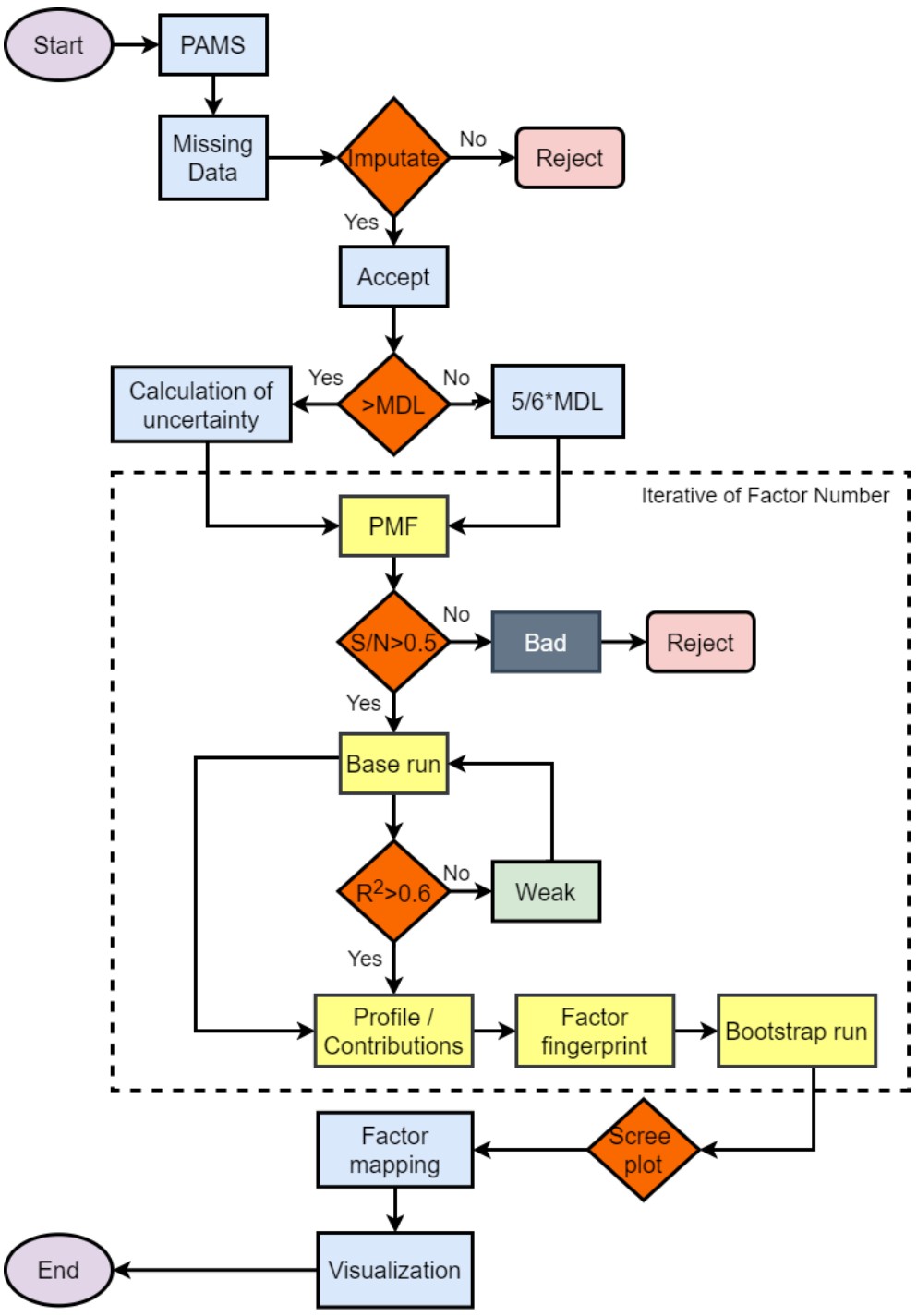

Figure 4: Overall framework of the Positive Matrix Factorization (PMF) model methodology. Processes in the dashed-line box are repeated for 3-8 factor number combinations. MDL: minimum detection limit.

Table 2: Categorization of PAMS-VOC species for PMF model source apportionment analysis. Grey-highlighted species represents unused species with abundant missing data >55% below minimum detection limit (MDL). Poor category species are identified for low S/N <0.5. Weak (Strong) category species are identified for S/N $\geq 0.5$ and $R^2 < 0.6$ ($R^2 \geq 0.6$) between measured and modelled concentration predicted by PMF model. Both unused and bad species are removed from PMF model analysis.

| No | VOC Species | Chaozhou | | Qiaotou | | Xiaogang | |
|---|---|---|---|---|---|---|---|
| | | MDL | Category | MDL | Category | MDL | Category |
| 1 | Ethane | Yes | Strong | Yes | Weak | Yes | Strong |
| 2 | Ethylene | Yes | Weak | Yes | Weak | Yes | Weak |
| 3 | Propane | Yes | Strong | Yes | Strong | Yes | Weak |
| 4 | Propylene | Yes | Weak | Yes | Weak | Yes | Weak |
| 5 | Isobutane | Yes | Strong | Yes | Weak | Yes | Strong |
| 6 | n-Butane | Yes | Strong | Yes | Weak | Yes | Strong |
| 7 | Acetylene | Yes | Strong | Yes | Weak | Yes | Strong |
| 8 | t-2-Butene | Yes | Poor | Yes | Poor | Yes | Poor |
| 9 | 1-Butene | Yes | Poor | Yes | Poor | Yes | Poor |
| 10 | cis-2-Butene | No | - | No | - | No | - |
| 11 | Cyclopentane | Yes | Poor | Yes | Poor | Yes | Poor |
| 12 | Isopentane | Yes | Strong | Yes | Weak | Yes | Strong |
| 13 | n-Pentane | Yes | Strong | Yes | Weak | Yes | Weak |
| 14 | t-2-Pentene | No | - | No | - | No | - |
| 15 | 1-Pentene | No | - | No | - | No | - |
| 16 | cis-2-Pentene | No | - | No | - | No | - |
| 17 | 2,2-Dimethylbutane | No | - | No | - | No | - |
| 18 | 2,3-Dimethylbutane | Yes | Poor | Yes | Poor | Yes | Poor |
| 19 | 2-Methylpentane | Yes | Poor | Yes | Poor | Yes | Poor |
| 20 | 3-Methylpentane | Yes | Poor | Yes | Poor | Yes | Poor |
| 21 | Isoprene | Yes | Strong | Yes | Strong | Yes | Strong |
| 22 | n-Hexane | Yes | Weak | Yes | Weak | Yes | Weak |
| 23 | Methylcyclopentane | Yes | Poor | Yes | Poor | Yes | Poor |
| 24 | 2,4-Dimethylpentane | No | - | No | - | No | - |
| 25 | Benzene | Yes | Weak | Yes | Weak | Yes | Weak |
| 26 | Cyclohexane | Yes | Weak | Yes | Weak | Yes | Weak |
| 27 | 2-Methylhexane | Yes | Poor | Yes | Poor | Yes | Poor |

| 28 | 2,3-Dimethylpentane | No | - | No | - | No | - |
|----|---------------------|-----|------|-----|--------|-----|--------|
| 29 | 3-Methylheptane | No | - | No | - | No | - |
| 30 | 2,2,4-Trimethylpentane | Yes | Strong | Yes | Weak | Yes | Strong |
| 31 | n-Heptane | Yes | Poor | Yes | Poor | Yes | Poor |
| 32 | Methylcyclohexane | Yes | Poor | Yes | Poor | Yes | Poor |
| 33 | 2,3,4-Trimethylpentane | Yes | Poor | Yes | Poor | Yes | Poor |
| 34 | Toluene | Yes | Weak | Yes | Strong | Yes | Strong |
| 35 | 2-Methylheptane | No | - | No | - | No | - |
| 36 | 3-Methylhexane | Yes | Poor | Yes | Poor | Yes | Poor |
| 37 | n-Octane | Yes | Poor | Yes | Poor | Yes | Poor |
| 38 | Ethylbenzene | Yes | Strong | Yes | Strong | Yes | Weak |
| 39 | m,p-Xylene | Yes | Strong | Yes | Strong | Yes | Weak |
| 40 | Styrene | No | - | No | - | No | - |
| 41 | o-Xylene | Yes | Strong | Yes | Strong | Yes | Weak |
| 42 | n-Nonane | No | - | No | - | No | - |
| 43 | Isopropylbenzene | Yes | Weak | Yes | Weak | Yes | Weak |
| 44 | n-Propylbenzene | Yes | Poor | Yes | Poor | Yes | Poor |
| 45 | m-Ethyltoluene | Yes | Weak | Yes | Strong | Yes | Strong |
| 46 | p-Ethyltoluene | Yes | Poor | Yes | Poor | Yes | Poor |
| 47 | 1,3,5-Trimethylbenzene | No | - | No | - | No | - |
| 48 | o-Ethyltoluene | Yes | Poor | Yes | Poor | Yes | Poor |
| 49 | 1,2,4-Trimethylbenzene | Yes | Weak | Yes | Strong | Yes | Weak |
| 50 | n-Decane | No | - | No | - | No | - |
| 51 | 1,2,3-Trimethylbenzene | Yes | Poor | Yes | Poor | Yes | Poor |
| 52 | m-Diethylbenzene | No | - | No | - | No | - |
| 53 | p-Diethylbenzene | No | - | No | - | No | - |
| 54 | n-Undecane | No | - | No | - | No | - |


## 3. Results & Discussions

### 3.1 Decomposition of Ozone Response

First-order sensitivity coefficients indicate the linear response of ozone concentrations to small changes in emissions. Ozone

response to small changes of daytime (09-15 LST) $NO_x$ emissions is dominated by negative sensitivities (VOC-limited) in

the urban area and positive sensitivities ($NO_x$-limited) in the inland area near the mountainous region (Figure 5(a)). Small

areas of extreme negative sensitivity were concentrated near the most intense $NO_x$ emissions in XG (magenta borderline in

Figure 5), a heavy industrial park district in the study domain. Although high VOC emissions are also present in the industrial park, the magnitude of first-order sensitivity to $NO_x$ emissions (-31 ppb $h^{-1}$) is relatively larger than that of VOC emissions (+18 ppb $h^{-1}$), indicating that $O_3$ linear response to $NO_x$ emissions is proportionally greater than that of VOC emissions from the industrial park (see Figure 5(a, b)). Moving away from the source emissions, the negative sensitivity to daytime $NO_x$ emission gradually increases and becomes positive in inland area, indicating the shift of ozone production sensitivity from a VOC-limited to $NO_x$-limited regime. The negative sensitivities to daytime $NO_x$ emissions extend to the coastal Pingtung county (green borderline in Figure 5), reflecting the downwind transport of $NO_x$ from the source region by the steering of northeasterly winds to westerly winds from the terrain effect and local circulation. Areas of positive sensitivity to VOC emissions overlap with areas of negative sensitivity to $NO_x$ emissions, confirming the VOC-limited regime, but also extend to $NO_x$-saturated areas. The larger area of VOC sensitivities could also be explained by the longer atmospheric lifetime of VOCs (hours to days for high OFP species such as toluene, ethylbenzene, xylene, ethane, and ethylene) as compared to $NO_x$ (1.0-4.5 hours) (Laughner and Cohen, 2019).

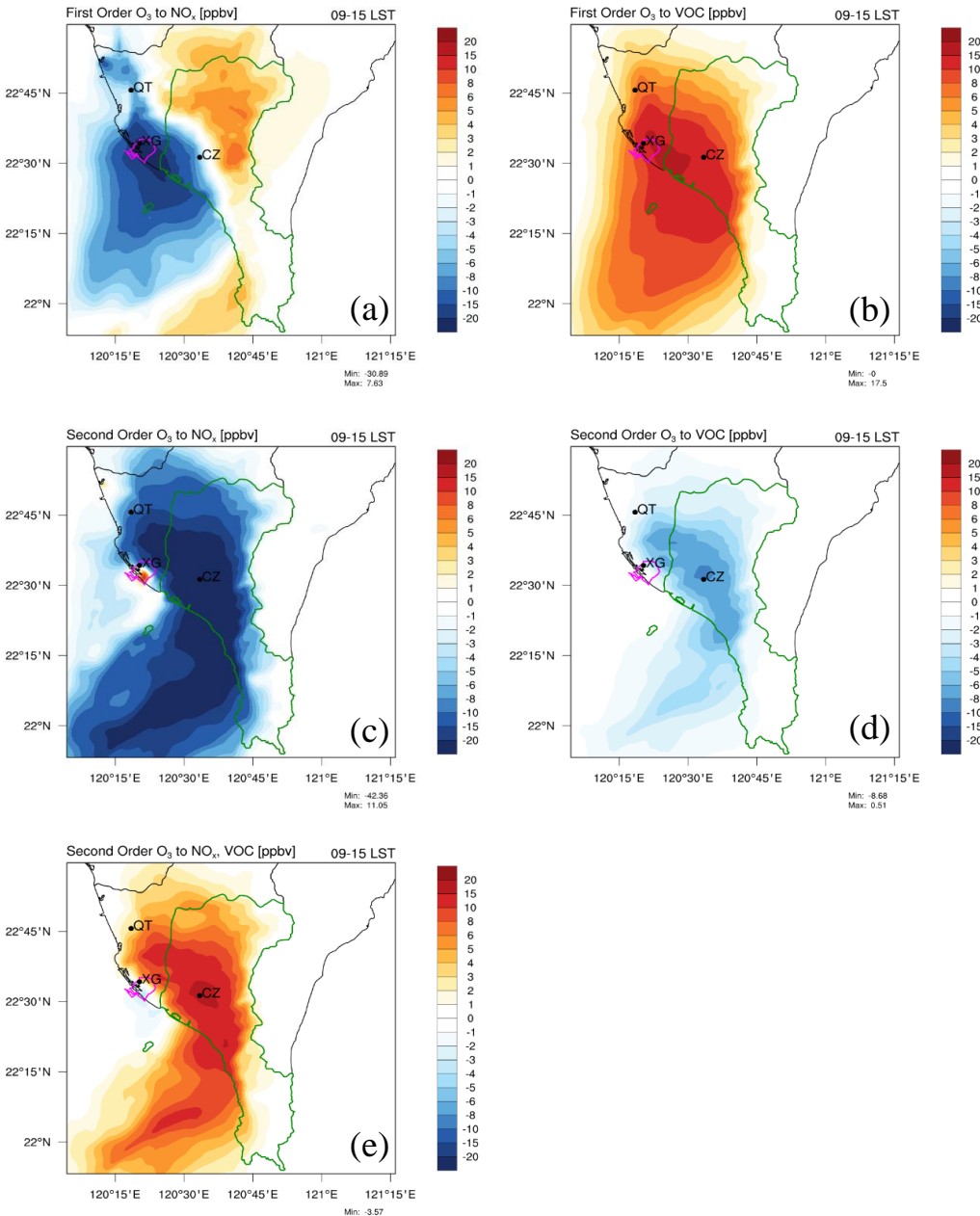

Figure 5: CMAQ-HDDM first-order sensitivity coefficient of $O_3$ to (a) $NO_x$ emissions, (b) VOC emissions, second-order sensitivity coefficient of $O_3$ to (c) $NO_x$ emissions, (d) VOC emissions, (e) second-order cross sensitivity coefficients of $O_3$ to $NO_x$, VOC emissions, at daytime 09-15 LST averaged during the entire simulation period. Magenta and green highlighted borderline represents Xiaogang District and Pingtung region, respectively.

Second-order sensitivity indicates the responsiveness of ozone concentration to broader changes in emissions and also delineates the non-linear sensitivities of ozone concentration to emissions. Most of the urban areas (except XG, an industrial park) exhibited negative second-order sensitivities to daytime $NO_x$ emissions, which when coincident with negative first-order sensitivities yields a negative $O_3$ convex response: urban $O_3$ increases linearly but decreases non-linearly with decreasing $NO_x$ emissions. Thus, in these areas, the large negative second-order sensitivities countered the linear negative

sensitivity of $O_3$ creating urban $O_3$ production conditions that were less sensitive than otherwise to changes in $NO_x$ emissions. In contrast, positive second-order sensitivities to daytime $NO_x$ emissions in XG were coincident with negative first-order sensitivities, which yielded a negative $O_3$ concave response with linear and nonlinear $O_3$ increases during decreasing $NO_x$ emissions (Fig. S2). This concave $O_3$ response is also observed in the Chicago O'Hare (ORD) area, which is highly polluted and severely impacted by aircraft emissions (Arter and Arunachalam, 2021); it is similar to XG where it also

hosted an international airport and several industrial parks. In XG, the $NO_x$ titration effect that consumes ozone is relatively higher than other urban areas of Kaoping due to the high $NO_x$/VOC condition (see Figure S6) and hence a rapid reduction in $NO_x$ level could further suppress the titration effect and increase the ozone formation rate. While for urban areas outside of XG, rapid $NO_x$ reductions push the condition into transition or $NO_x$-limited regime and suppression of the titration is countered by less efficient $O_3$ production. Cross sensitivities indicate the interaction effects of the sensitivities when both

$NO_x$ and VOC emissions are changed simultaneously (limited to $NO_x$ and VOC changes in the same direction, positive or negative). In most cases, decreasing the $NO_x$ (VOC) emissions enhanced the sensitivities of ozone to the limiting precursors, $NO_x$ (VOC) due to the positive cross-sensitivity of ozone to $NO_x$ and VOCs (see Figure 5e). Cross-sensitivities also indicate how results from emission control strategies may change from summing the results of decreasing individual $NO_x$ and VOC emissions (Arter and Arunachalam, 2021). Cross-sensitivities usually have only a small contribution in highly VOC-limited

or highly $NO_x$-limited conditions because the associated net ozone production rates are driven singly either by VOC or $NO_x$, respectively (Sillman, 1999; Cohan et al., 2005). This result is reproduced in our model simulation; low cross-sensitivities between $NO_x$ and VOC are simulated in XG (highly VOC-limited condition) as well as remote areas in west side of the mountain range in the southern Taiwan (highly $NO_x$-limited condition), while high positive cross-sensitivities between $NO_x$

and VOC emissions are concentrated in downwind suburban areas (e.g. Pingtung county) (Figure 5e). The latter result

highlights a potential under-prediction of $O_3$ formation sensitivity if the cross-sensitivity term is excluded in suburban areas.

### 3.2 Taylor-series Expansion Approximation

We then sought to exploit the sensitivity coefficients determined above to estimate the impact of $NO_x$ and VOC emission

reductions on ozone concentrations in southern Taiwan from the time of our study period, 2018 to 2025. To determine the

level of emissions reduction in the near term, we referred to the long-term trend of projected emissions in Taiwan as reported

in the Taiwan Emission Data System (TEDS 11.0; https://air.epa.gov.tw), which estimated an overall reduction of 53%

($NO_x$) and 14% (NMHC) over 22 years (2007-2028). In recent years, the decreasing trends for both $NO_x$ and NMHC

emissions have slowed down (see Figure S7). We then used 5% emission reductions over the 8-year near-term period from

2018-2025 as a conservative estimate. Although this reduction is small, the Taylor-series approximation (Eq. 4-5) considers

the first-order, second-order and cross sensitivity coefficients of $O_3$ to this change in $NO_x$ and VOC emissions and provides a

clearer picture on how $O_3$ will change with a parallel reduction in emissions in the near future.

We estimated the ozone concentration at different scenarios represented by different emission control strategies for the

typical seasonal transition period already simulated above, which often causes high $O_3$ episodes in the study area. For

simplicity, reduction of domain-wide $NO_x$ and VOC emissions is considered. Given the high nonlinearity of the ozone

response to emissions of $NO_x$ and VOC, the Taylor Series approximation is necessary to consider the higher order terms

involved in second-order and cross-term sensitivity of $NO_x$ and VOC emissions. To compare a series of targeted emission

control strategies, four experiments are considered: (1) baseline, (2) $NO_x$ control, (3) VOC control, and (4) $NO_x$ & VOC

control. All experiments reduce the targeted emission by 5% except for the baseline experiment representing the current

emission with no perturbations. Figure 6 shows the baseline $O_3$ concentration and its corresponding response to $NO_x$ control,

VOC control, and $NO_x$ & VOC control at daytime 12 LST. Benefits of $NO_x$ control (i.e. reduction of $O_3$ by 0.6-0.8 ppbv) are

seen to dominate over the inland area. However, an adverse effect of increased $O_3$ concentration >1.8 ppbv from $NO_x$ control

is simulated near the western coastal urban area where high $NO_x$ emissions are concentrated. This agrees with our previous

analysis that urban $O_3$ concentrations are estimated to increase from decreasing $NO_x$ emissions for most of the urban areas that follow $O_3$ concave sensitivity behaviors and increase more for highly polluted urban areas (i.e. XG) that follows $O_3$ convex sensitivity behaviors. In contrast, no adverse effect is observed from VOC control, but rather resulted in decreased $O_3$ concentration over much of the study region. The largest ozone response to VOC control is -1.4 ppbv in the urban area, but also extended to the inland area. Control of both $NO_x$ and VOC emissions resulted in reduced inland $O_3$ concentration, a marginal difference in urban areas, and slightly increased $O_3$ concentration near the large point source area in XG.

The decreasing trend of $NO_x$ emission in the future is likely to continue owing to adoption of electric vehicles (EV), which may impact the urban and inland areas differently due to the respective sensitivity regime. Using the Taylor-series approximation, we estimated $\partial O_3$ against $\partial VOC$ emissions at different levels of projected $\partial NO_x$ emissions for Xiaogang, XG (urban) and Chaozhou, CZ (inland) at 12:00 LST (Figure 7). Projected $\partial NO_x$ emissions at 0% (black line) and -5% (red line) represents the current year 2018 and near future 2025, respectively while other projected $\partial NO_x$ emissions at arbitrary -50% (blue), -25% (magenta), and +25% (green) are presented for alternative scenarios. At CZ, our results show that inland $O_3$ becomes less sensitive to varying $\partial VOC$ emissions (i.e. less benefits of VOC control) at greater reductions of $\partial NO_x$ emission at -25% and -50% (i.e. moving into the $NO_x$-limited regime). On the other hand, $\partial O_3$ at XG is always sensitive to varying levels of $\partial VOC$ emissions at all levels of $\partial NO_x$ scenarios and $\partial O_3$ increases in parallel following the $\partial NO_x$ order -50%, -25% and -5%. This is expected in a highly polluted area that favors VOC-limited conditions where $O_3$ could have the reverse effect to $NO_x$ due to the reduced titration effect for decreasing $NO_x$. Besides, at all projected levels of $\partial NO_x$ emissions, urban $O_3$ continues to decrease with decreasing VOC emissions. We showed in the Taylor-series approximation analysis when we hypothetically reduce $NO_x$ emission at arbitrary -50% and -25% scenario, which should expect an improvement in the overestimation of $NO_2$, urban $O_3$ at XG remains in a VOC-limited condition. Therefore, it is certain that the VOC-limited condition at urban areas of southern Taiwan is mainly due to the high anthropogenic $NO_x$ emissions and VOC controls are beneficial in reducing $O_3$ concentration in highly polluted urban areas.

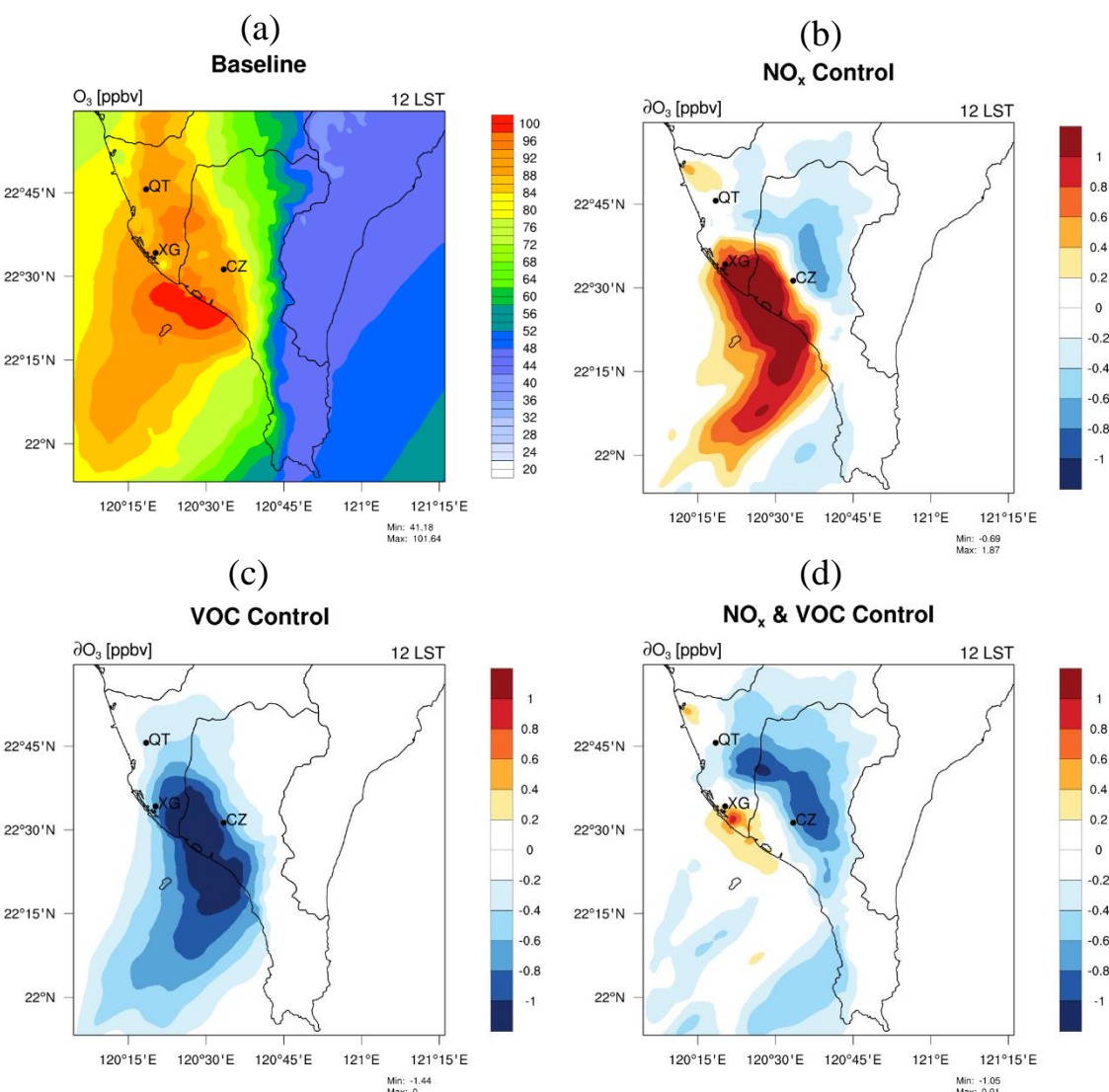

Figure 6: Spatial distribution of $O_3$ concentration in (a) baseline with no perturbations in $NO_x$ and VOC emissions, and
changes in $O_3$ concentration under (b) $NO_x$ control scenario, (c) VOC control scenario, and (d) $NO_x$ & VOC control scenario
at daytime 12 LST. All scenarios reduced the targeted emissions by 5% except for the baseline.

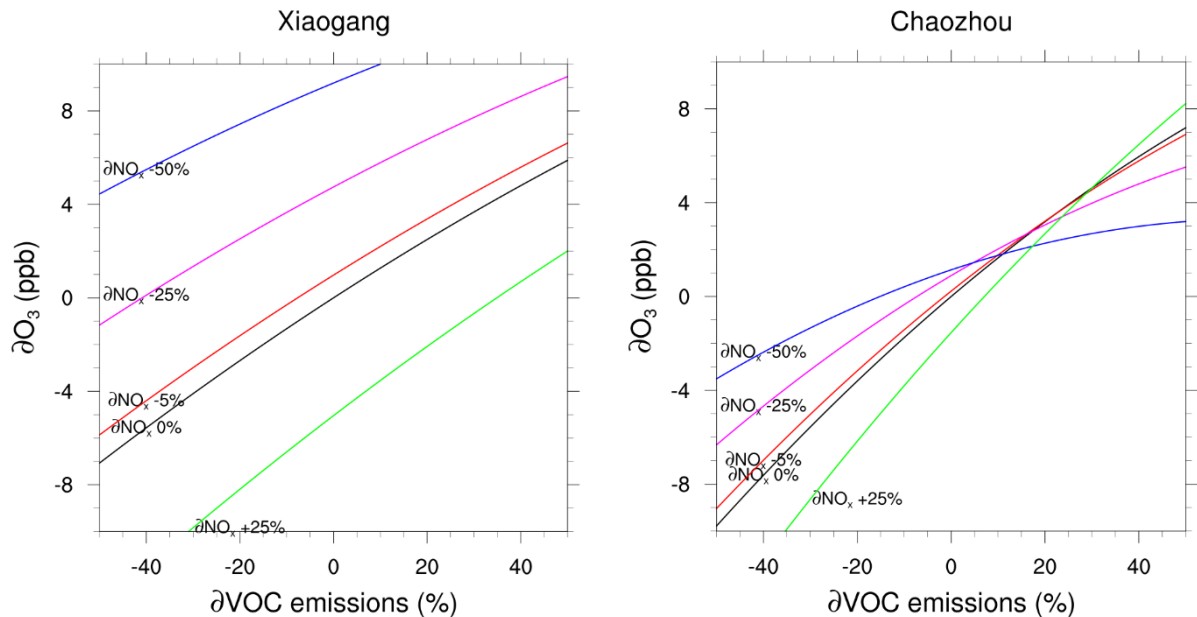

Figure 7: Taylor-series approximation of $\partial O_3$ against $\partial VOC$ emissions at different level of projected $\partial NO_x$ emissions for Xiaogang (urban) and Chaozhou (inland) averaged at 12:00 LST. Projected $\partial NO_x$ emissions at 0% (black line) and -5% (red line) represents the current year 2018 and near future 2025, respectively; other projected $\partial NO_x$ emissions at arbitrary -50%, ±25% are presented for far-future comparison purpose.

**3.3 Sensitivity of Individual Modeled VOC Species**

In this subsection, we further evaluate the sensitivity of the ozone response to individual modeled VOC emissions using the CMAQ direct decoupled method (DDM-3D). Note that only the first-order sensitivity coefficient of individual VOC species in CB6 is calculated considering the high computational cost and the second-order sensitivity coefficients of ozone to total VOC emissions have a low contribution. A total of 20 individual sensitivity tests were performed to target each modeled lumped VOC class or individual species (refer to Table 1 for a complete list of modeled VOC species). Figure 8(a) shows the daytime CMAQ DDM sensitivity coefficient of ozone to the VOC emissions, arranged in ascending order and separated for urban and inland area. The urban area sensitivity coefficients are substantially higher than that in the inland area for all species, which is expected due to the VOC-limited regime in the urban area. Among the 20 VOC species, the six most

important VOC species were identified as XYL (xylene), OLE (terminal C-C olefins), PAR (paraffins), ETH (ethene), TOL (toluene), IOL (internal C-C olefins).


XYL and TOL are aromatic hydrocarbons that are included in the BTEX group, along with benzene and ethylbenzene. The ranking of BTEX with respect to ozone formation potential in a high $NO_x$ environment is often done using maximum incremental reactivity (MIR), which uses unitless coefficients and indicates the amount contributed to ozone formation in the air mass by individual compounds. Based on the MIR scale adopted from the literature (Atkinson, 1997; Carter, 1994), m,p-

xylene (8.20) are the most dominant contributors to ozone formation among BTEX, followed by toluene (2.70), ethylbenzene (2.70) and benzene (0.42). Our result agrees with the MIR scale, which yielded classifications of XYL and TOL as the second and fifth most sensitive VOC species to $O_3$ among the 20 VOC species considered in the sensitivity tests. Other sensitive VOC species to $O_3$ formation include alkanes (i.e. PAR (MIR = 0.32)) and alkenes (i.e. OLE (8.24), ETH (4.37), IOL (13.11)).


To gain additional insights on the ozone response to the six most sensitive VOC species, we grouped them into three categories (i.e. alkenes: OLE, ETH, IOL; aromatics: TOL, XYL; alkanes: PAR) and plotted the spatial distributions of the sensitivities at 12 LST (Figure 8(b-d)). The sensitivity of ozone to alkenes and alkanes have a similar pattern with higher sensitivity concentrated near the emission source and slightly shifted towards inland/mountainous and Pingtung county due

to the local circulation and the prevailing northeasterly winds. For the sensitivity of ozone to aromatics, the spatial distribution has a smaller area coverage and higher sensitivity is more concentrated near the emission source. To render a clearer picture on the spatial distribution of $O_3$ sensitivity to each VOCs component, the spatial distribution of some highly sensitive VOC component emissions (i.e. XYL, OLE, PAR, ETH, TOL, IOL) are provided in the supplementary material (see Figure S8). The role of land-sea breeze circulation in transporting the local $O_3$ precursors (i.e. VOC) from urban to

inland area is well reflected by the diurnal profile of the sensitivity coefficients (Figure 9). An obvious shift is simulated in inland area as compared to the urban area; the urban sensitivity coefficients peak at 10 LST while the peak in inland area is at 13 LST. A lag of 2-3 hours delay is mainly attributed to the daytime sea-breeze penetration that pushes the urban polluted

air towards the inland area due to the differential heating between the land and sea surfaces during the noon hours. This result also tells us that VOC emissions from urban area are largely responsible for the inland high ozone episodes.


Using an approach similar to OFP, we quantified the relative ground-level ozone impacts of the six most sensitive VOC species by multiplying the MIR coefficients to its corresponding concentrations at 12 LST (see Figure 8(e-g)). Due to the abundance of PAR, OFP calculated for alkanes emission is significant over the western coastal region of southern Taiwan. The longer atmospheric lifetime of alkanes makes it more susceptible to transportation and impacting a wider area. Despite

alkanes having the lowest sensitivity coefficient of the six groups of VOCs in the study, the significance of alkane OFP extends over a larger area, highlighting the need for controlling alkane-related emissions to reduce the ozone pollution problem over the study area. OFP of alkenes features a point source distribution pattern indicating it is more likely related to large point source emissions. OFP of aromatics has the smallest area coverage and is highly concentrated near the populated industrial park of the study area in southern Taiwan. Among the three main contributors, aromatics have the highest maxima

OFP value 56 ppbv, followed by alkenes (49 ppbv) and alkanes (40 ppbv).

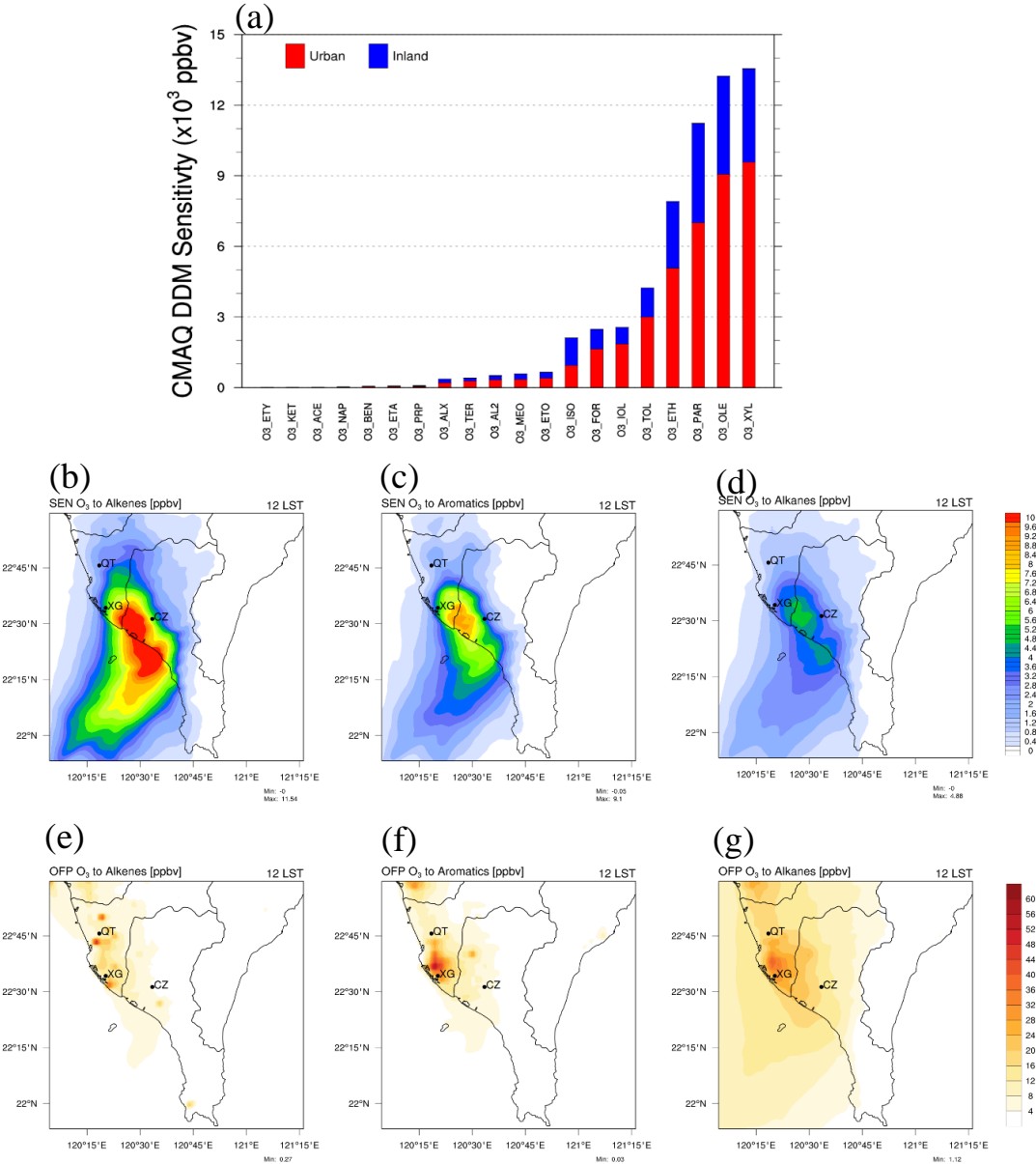

Figure 8: (a) Daily averaged CMAQ-DDM first-order sensitivity coefficient of $O_3$ concentrations calculated per number of grids to each modelled VOC species arranged in ascending order for urban and inland area. Sensitivity of $O_3$ and ozone formation potential (OFP) to (b,e) alkenes emissions (OLE + ETH + IOL), (c,f) aromatics emissions (XYL + TOL), and (d,g) alkanes emissions (PAR) at 12 LST.

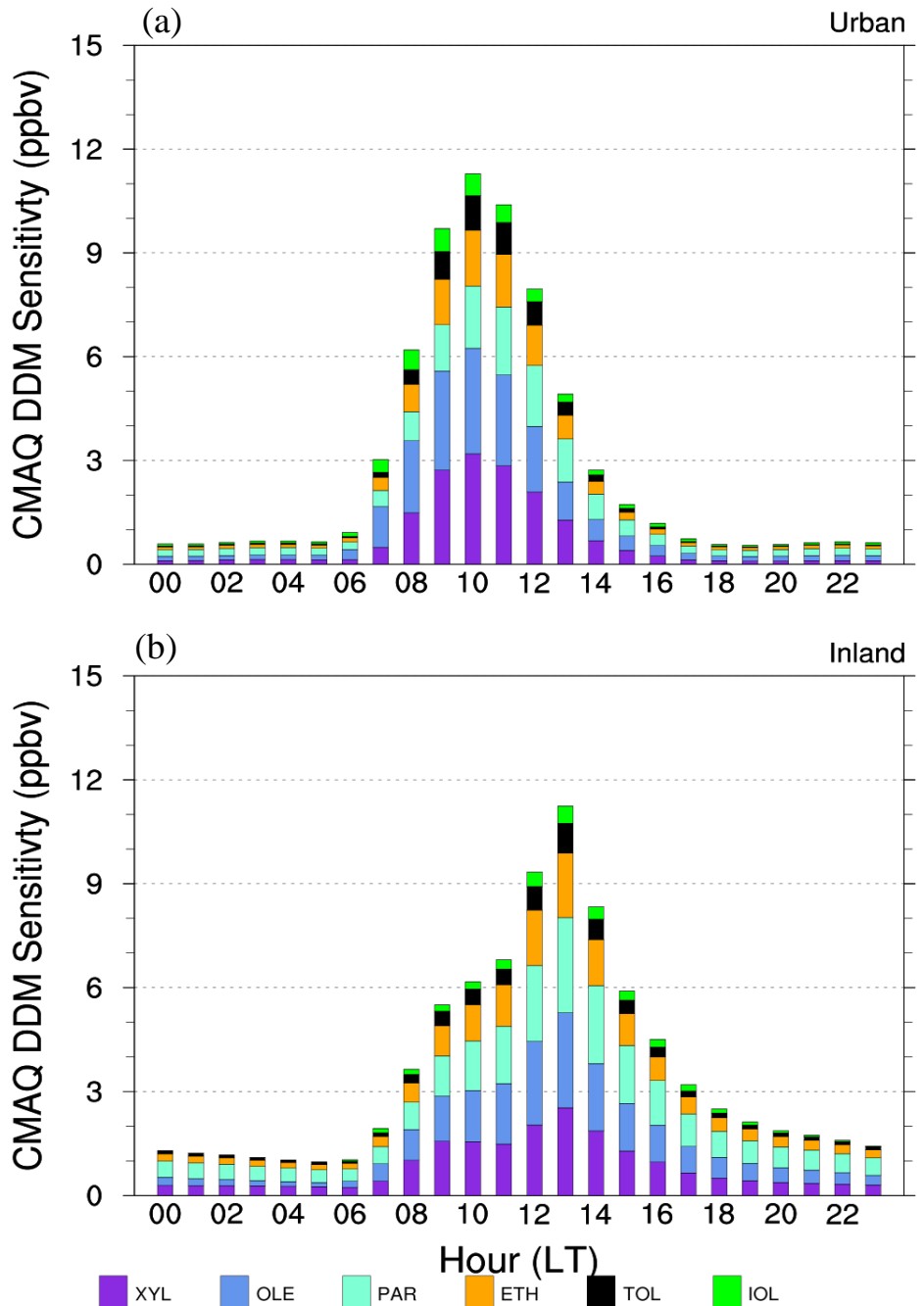

Figure 9: Diurnal variations of the CMAQ-DDM sensitivity of O3 to the six most sensitive VOC modelled species (i.e. terminal olefins, OLE; xylene, XYL; paraffin, PAR; ethene, ETH; toluene; TOL; internal olefins, IOL) averaged for (a) urban and (b) inland area, during the entire simulation period.


Figure 10 shows the relative sensitivities for each modelled VOC to $O_3$ (DDM – Figure 10(a)) and its corresponding emission rate (EMIS – Figure 10(b)) averaged in urban and inland areas during the entire simulation period. Similar to the above analysis, the three main contributors were identified as alkenes (42.2% for urban; 37.1% for inland), aromatics (29.1%; 25.1%), and alkanes (18.2%; 23.4%). Other less important contributors were formaldehyde (4.1%; 3.9%), isoprene (2.3%; 5.4%), alcohols (1.9%; 2.6%) and aldehydes (0.9%; 1.6%). Due to the abundant biogenic emissions surrounding the inland area, inland ozone is more sensitive to isoprene (5.4%) as compared to the urban area. Besides, inland ozone is also more sensitive to alcohols (MEOH + ETOH) and aldehydes (ALD2 + ALDX) when compared to the urban area. This is also consistent with the higher contributions of alcohols (10.0%) and aldehydes (1.4%) inland than in urban areas (4.3% and 0.8%, respectively) (Figure 10(b)).

Overall, the DDM sensitivity tests identified three main speciated VOC groups contributing to ozone formation, namely alkenes, aromatics, and alkanes. The six most sensitive VOC surrogates in CB6 to ozone formation in the study area are XYL, OLE, PAR, ETH, TOL, IOL in descending order. Sources of these targeted VOC species are highly variable and previous studies have attributed these species to a wide-range of emission sources not limited to traffic, industrial, petrochemical, manufacturing, solvent usage, etc. Therefore, in order to identify the sources of these VOC species for better emission control policy, source apportionment is necessary.


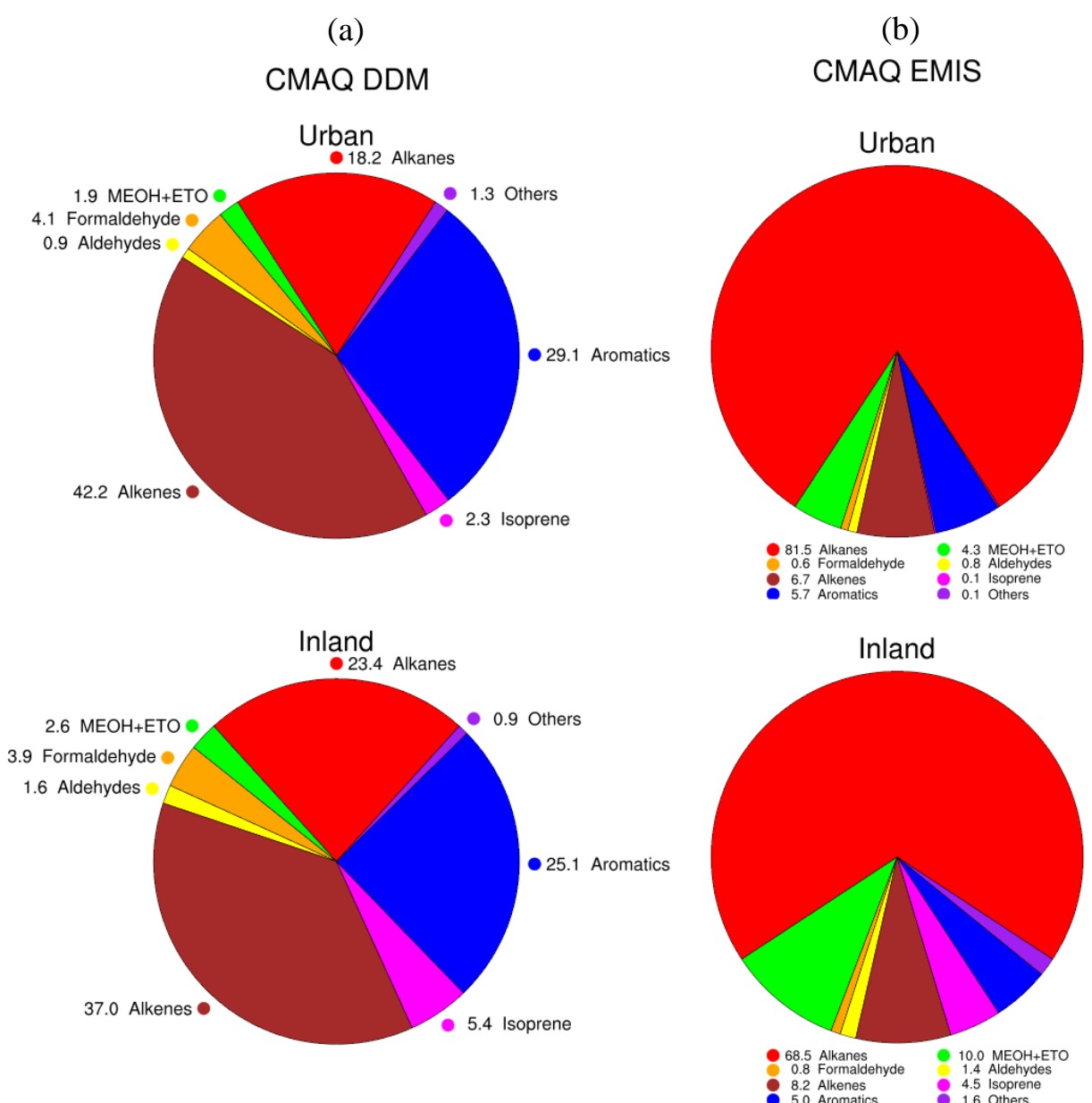

Figure 10: Relative distribution in percentage of (a) CMAQ DDM $O_3$ sensitivity and (b) CMAQ EMIS hydrocarbon emissions to individual VOC summed across different groups averaged for urban and inland area during the entire simulation period.

**3.4 Descriptive Statistics of PAMS-VOC Data & PMF Optimal Solution**

In this subsection, we first describe the statistics of the PAMS-VOC data measured at Chaozhou (CZ), Qiaotou (QT) and Xiaogang (XG) which were used to drive the PMF source apportionment. CZ station is an inland station located far away from the urban core while QT and XG are urban stations. The top 15 major VOC species measured at the three PAMS stations are summarized in Table 3 and other species are shown in Figure S9. These top 15 major VOC species accounted for 65.9%, 70.7% and 68.7% of the total VOC concentrations in CZ, QT and XG, respectively. Overall, aromatics contributed the largest proportion, accounting for 117.3 ppbv (43.5%) in CZ, 312.2 ppbv (53.9%) in QT, and 361.7 ppbv (51.2%) in XG. Toluene was the most abundant VOC species in all three stations, accounting for $46.3 \pm 45.0$ ppbv (17.2% of total) in CZ, $140.4 \pm 96.2$ ppbv (24.3%) in QT, and $143.0 \pm 97.7$ ppbv (20.2%) in XG. Toluene is also commonly used in chemical industry (Mo et al., 2015), iron and steel industry (Nogueira et al., 2011), fuel evaporation (Wu et al., 2016), organic solvent usage (Chen et al., 2019) and a by-product of vehicle exhaust (Dai et al., 2013). The second largest contributor to total VOC concentrations were alkanes, accounting for 42.7% in CZ, 40.0% in QT, and 40.6% in XG. Among the alkanes, n-butane, propane and isopentane comprised the highest proportions, each reaching around 5% of the total VOC concentrations. n-butane isomers are emitted from a variety of sources including petroleum production and natural gas emissions (Rossabi and Helmig, 2018). In the alkenes category, ethylene and propylene are major contributors which are commonly used as raw materials in petrochemical industry and refineries. Among the three stations, the highest proportion of alkenes group is observed in XG, accounting for 7.0% of total VOCs concentrations, with 3.6% attributed to propylene.

Table 3: Concentration (mean ± std) and proportions (%) of the top 15 PAMS-VOC species in ascending order at CZ, QT, and XG during 1-31 October 2018. Bold/italic species represents the unique species that present in the top 15 at each station. All units in ppb.

| Chaozhou Species | mean | std | % | Qiaotou Species | mean | std | % | Xiaogang Species | mean | std | % |
|---|---|---|---|---|---|---|---|---|---|---|---|
| Toluene | 46.28 | 44.96 | 17.2 | Toluene | 140.39 | 96.21 | 24.3 | Toluene | 142.95 | 97.65 | 20.2 |
| *Acetylene* | 18.94 | 51.23 | 7.0 | m,p-Xylene | 49.72 | 28.97 | 8.6 | m,p-Xylene | 63.77 | 52.17 | 9.0 |
| n-Butane | 14.47 | 9.18 | 5.4 | n-Butane | 31.67 | 18.18 | 5.5 | Isopentane | 36.04 | 24.32 | 5.1 |
| m,p-Xylene | 13.56 | 11.64 | 5.0 | Isopentane | 26.81 | 16.41 | 4.6 | n-Butane | 33.79 | 19.91 | 4.8 |
| Propane | 13.52 | 7.72 | 5.0 | Propane | 23.29 | 10.88 | 4.0 | Propane | 26.09 | 18.18 | 3.7 |
| Isopentane | 11.25 | 6.38 | 4.2 | Ethylbenzene | 17.87 | 9.61 | 3.1 | *Propylene* | 25.51 | 105.81 | 3.6 |
| Benzene | 8.94 | 6.27 | 3.3 | 1,2,4-Trimethylbenzene | 17.25 | 10.59 | 3.0 | o-Xylene | 23.29 | 18.34 | 3.3 |
| Ethane | 8.03 | 2.61 | 3.0 | o-Xylene | 17.18 | 10.18 | 3.0 | Ethylbenzene | 21.87 | 17.96 | 3.1 |
| *Isopropylbenzene* | 7.69 | 10.49 | 2.9 | Benzene | 16.96 | 9.67 | 2.9 | 1,2,4-Trimethylbenzene | 20.01 | 15.99 | 2.8 |
| Isobutane | 7.30 | 4.23 | 2.7 | Isobutane | 14.97 | 7.67 | 2.6 | Benzene | 18.84 | 14.47 | 2.7 |
| 1,2,4-Trimethylbenzene | 6.27 | 8.38 | 2.3 | n-Pentane | 11.71 | 11.72 | 2.0 | n-Pentane | 18.32 | 20.14 | 2.6 |
| n-Pentane | 5.66 | 3.81 | 2.1 | Ethane | 10.86 | 4.02 | 1.9 | Isobutane | 15.44 | 8.47 | 2.2 |
| Ethylbenzene | 5.44 | 3.98 | 2.0 | m-Ethyltoluene | 10.27 | 7.34 | 1.8 | *Cyclohexane* | 14.32 | 26.41 | 2.0 |
| o-Xylene | 5.20 | 4.28 | 1.9 | *n-Hexane* | 10.17 | 6.97 | 1.8 | m-Ethyltoluene | 12.85 | 8.76 | 1.8 |
| *Isoprene* | 5.14 | 5.02 | 1.9 | *2,2,4-Trimethylpentane* | 10.15 | 5.08 | 1.8 | *Styrene* | 12.30 | 12.77 | 1.7 |
| **Alkanes** | **114.96** | | **42.7** | **231.42** | | **40.0** | **286.61** | | **40.6** |
| **Alkenes** | **18.33** | | **6.8** | **31.41** | | **5.4** | **49.37** | | **7.0** |
| **Alkynes** | **18.94** | | **7.0** | **3.88** | | **0.7** | **9.00** | | **1.3** |
| **Aromatics** | **117.32** | | **43.5** | **312.18** | | **53.9** | **361.69** | | **51.2** |
| **Top 15** | **177.68** | | **65.9** | **409.28** | | **70.7** | **485.39** | | **68.7** |
| **Total VOCs** | **269.55** | | **100.0** | **578.89** | | **100.0** | **706.67** | | **100.0** |

Determining the number of factors is a critical step in receptor-based source apportionment methods like PMF analysis. Combinations of 3-8 sources were used to determine the optimum number of VOC sources for each PAMS station. We also calculated the uncorrelated bootstrap (BS) mapping to assist in identifying the optimum number of factors in each station. A high number of uncorrelated BS mapping indicates excessive factors are fitted in the model; therefore, this number should be kept as low as possible to avoid over-fitting. In CZ, the $Q_{true}/Q_{expected}$ was at a minimum in the 6-factor solution and the uncorrelated BS mapping was low (n=3), indicating this was the optimal solution for that site (Figure S10). Using the same protocol, the optimal solution was determined as 7 factors for QT and 5 factors for XG. Overall, a total of 8 unique factors were identified at the three stations: biogenic, solvent usage, vehicle emissions, plastic industry, manufacturing industry, mixed industry, aged air mass and motorcycle exhausts (Figure 11). The diurnal variations of the PMF factor contribution at Chaozhou, Qiaotou, and Xiaogang PAMS stations are also shown in Figure 12. Among the 8 resolved factor profiles, vehicle emissions (22%) were the largest contribution to total VOC concentrations, followed by mixed industry (21%), solvent usage

(17%), biogenic (12%), plastic industry (10%) and other factor profiles (i.e. aged air mass, motorcycle exhausts and manufacturing industry). The diurnal pattern of factor distribution of biogenic sector displays a clear diurnal cycle peak at the noontime 10-12 LST in all three stations. Meanwhile, the motor exhaust and vehicle emissions sector peak during the morning and evening traffic rush hours. The aged airmass sector is only identified in CZ and its hourly factor contribution

mostly existed at more stable levels when compared to other factors except for an obvious peak at 12 LST due to the sea-breeze penetration that pushes the urban polluted aged air mass towards inland area. The diurnal pattern related to industrial activity exhibits different peak hours depending on sector and station. For instance, the hourly factor contribution of solvent usage in XG has a clear bimodal peak at 10 and 16 LST; plastic industry in QT and CZ has a clear diurnal cycle peak in the noon time at 11 and 13 LST; mixed industry in QT and XG has a clear bimodal peak at 07 and 18 LST. Details of each

source profile and comparison with other PMF studies are discussed in Supplementary Material - Source Profiles of PMF Model.

**3.5 Dominant Sources of Highly Sensitive VOC Species**

We then mapped the CMAQ CB6 modeled VOC species to the PMF apportioned sources in order to identify the dominant

sources of highly sensitive VOC species (i.e. XYL, OLE, PAR, ETH, TOL, IOL). The dominant sources of alkene species (OLE, ETH, IOL) were identified using ethylene and propylene as tracers, sources of TOL were tracked by toluene and m-ethyltoluene, sources of XYL were tracked by all three xylene isomers, and sources of PAR were identified using isobutane, n-butane, isopentane, n-pentane, n-hexane and 2,2,4-trimethylpentane.

We also developed a composite index, $\psi$, to quantitatively combine the CMAQ-DDM sensitivity coefficient and PMF resolved factor contribution in order to identify the key VOC species for an effective ozone abatement strategy. The index was calculated according to the $j$-th species and $k$-th source:

$$\psi_{jk} = \frac{\partial C}{\partial \varepsilon_j} \times f_{jk} \tag{9}$$

where $C$ is the trace gas ($O_3$) concentration, $\varepsilon$ is the unitless scaling factor for $j$ emission (see Eq. 2-3), $f_{kj}$ is the $j$-th species concentration to the $k$-th source. The first term in Eq. (9) is obtained from the CMAQ-DDM calculated first-order sensitivity coefficient where a higher value denotes greater sensitivity (extreme low sensitivity <0.01 ppb is masked out); the second term is the PMF resolved factor contribution where higher values of a particular species, $j$, indicate greater apportionment to the source, $k$. Higher values of the composite index indicate a greater priority should be given to that source-species.

According to the PMF source apportionment (Figure 13), the dominant source of alkene-related emissions (OLE, IOL, ETH) was attributed to mixed industry (i.e. petrochemical industry, printing industry and metal industry) (Chen et al., 2022; Pinthong et al., 2022). This source profile contributed 61.5% propylene and 47.6% ethylene in XG; 52.4% propylene and 37.9% ethylene in QT. Meanwhile, propylene and ethylene in CZ were mainly attributed to vehicle emissions and aged air masses. Given that alkene-related emissions are closely linked to large point sources (see Figure 8(e)), control of alkene emissions should focus on the mixed industry, particularly petrochemical plants. Among the three stations, the highest composite index associated with ethylene and propylene summed across all factors was obtained at XG. Though ethylene and propylene have higher sensitivities compared to other species, their composite index ($\psi_{ETHY,\ all\ k}$ = 5.7 and $\psi_{PROP,\ all\ k}$ = 7.4 – see Figure 13) is relatively low due to their low-abundance 11.5 ppb (1.6% of total VOCs) and 25.5 ppb (3.6%), respectively, making them less important in terms of priority.

The highest composite index was assigned to aromatic compounds such as toluene ($\psi_{TOL,\ all\ k}$ = 12.1, 16.4, 53.5 in CZ, QT and XG, respectively), m,p-xylene ($\psi_{mp\text{-}XYL,\ all\ k}$ = 12.0, 19.7, 53.5), o-xylene ($\psi_{o\text{-}XYL,\ all\ k}$ = 4.5, 6.7, 19.0) due to their high-abundance toluene 46.3 – 143.0 ppb (17.2 – 24.3% of total VOCs), m,p-xylene 13.6 – 63.8 ppb (5.0 – 9.0%) and o-xylene 5.2 – 23.3 ppb (1.9 – 3.3%) in all three stations (see Figure 13 & Table 3). Although toluene has much higher abundance than xylene, the composite index for both species summed across all sources is similar because the sensitivity coefficient of xylene ($\partial C/\partial \varepsilon$ > 12 ppb) is relatively higher than toluene ($\partial C/\partial \varepsilon$ > 3 ppb) (see Figure 8(a)). This indicates that emission control of both of these compounds is of high priority considering their high-abundance and high sensitivity in ozone formation. The dominant sources of aromatic-related emissions (XYL, TOL) are mainly attributed to solvent usage and

mixed industry. Solvent usage, including building coatings, paint thinners, and other products thinners (Li et al., 2021; Chen et al., 2019; Wu et al., 2016; Huang and Hsieh, 2020), contributed 64.5% xylene and 32.2% toluene in CZ; 40.6% xylene and 29.7% toluene in QT; 38.5% xylene and 43.3% toluene in XG. Mixed industry contributed 26.2% xylene and 21.9% toluene in QT; 43.5% xylene and 45.3% toluene in XG. Vehicle emissions, including motorcycle exhaust contributed less than 10% of xylene and toluene in CZ and QT. Therefore, emission control of XYL and TOL should target on solvent usage particularly painting, coating, synthetic fragrances, adhesives and cleaning agents, and industrial sources in the region.

The dominant source of alkane-related emissions (i.e. PAR) is attributed to vehicle emissions in all three stations. The main tracers identified in the PMF source apportionment related to vehicle emissions are isobutane, n-butane, isopentane, n-pentane; these compounds accounted for 54.9%, 53.0%, 47.0%, and 47.0% of the PMF normalized factor contribution in CZ; 54.8%, 58.7%, 25.9%, and 29.1% in QT; 44.5%, 43.8%, 23.9%, and 26.8% in XG, respectively. Motorcycle exhaust also contributed to these compounds but the average contribution was relatively lower (e.g. <20% in QT). High amounts of C4-C5 alkanes (i.e. isobutane, n-butane, isopentane, n-pentane) are known indicators of traffic-related sources (Yu et al., 2021b; Huang and Hsieh, 2020; Chen et al., 2019). Among these compounds, n-butane had the highest composite index due to its high-abundance 15-34 ppb (5.1-5.5% of total VOCs). Other than freshly emitted vehicle emissions, aged air masses also contributed a significant amount (~30%) to these compounds at CZ, reflecting the role of land-sea breeze circulation transporting urban vehicle plumes inland. In addition, acetylene is a well-known indicator for combustion sources and related to liquefied petroleum gas (LPG) leakage. The ratio of propane, n-butane, or i-butane to acetylene is often used to assess the domination of the LPG-vehicle sources across different cities (An et al., 2014). These ratios were 0.7, 0.8, 0.4 in CZ, 6.0, 8.2, 3.9 in QT, and 2.9, 3.8, 1.7 in XG, respectively, which were much lower than 11.5, 1.8, 2.6 in Guangzhou (Zhang et al., 2013) and 11.4, 5.0, 2.3 in Mexico city (Blake and Rowland, 1995) that are heavily impacted by LPG-vehicle emissions. This indicates that there were relatively less LPG-fueled vehicles in southern Taiwan whereas LPG is merely used to fuel a gas stove for domestic household use or catering. Therefore, emission control of PAR species should focus on the gasoline-fueled vehicle emissions particularly in heavy-traffic cities.

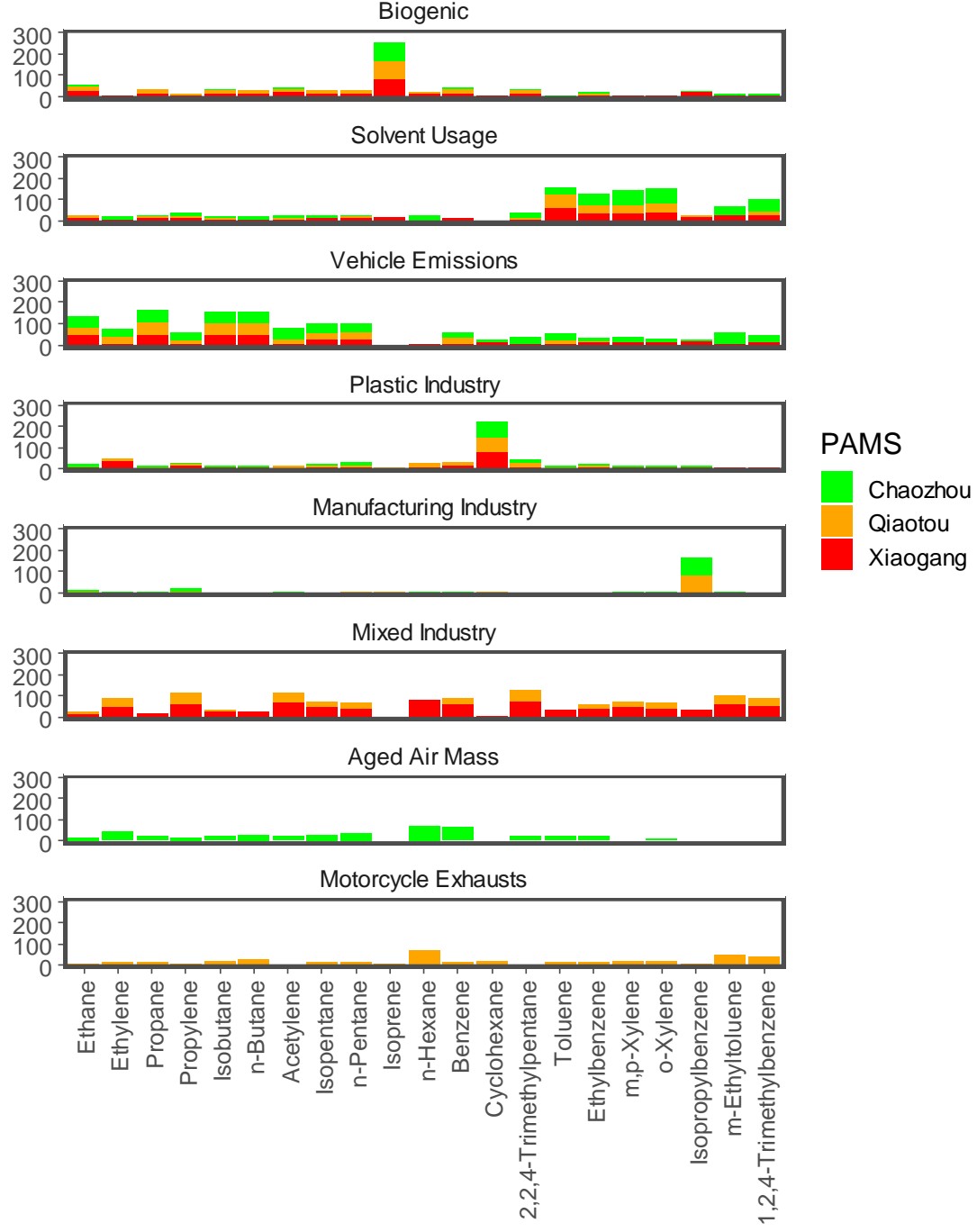


Figure 11: Source profiles calculated using the PMF model at Chaozhou, Qiaotou, Xiaogang PAMS stations.

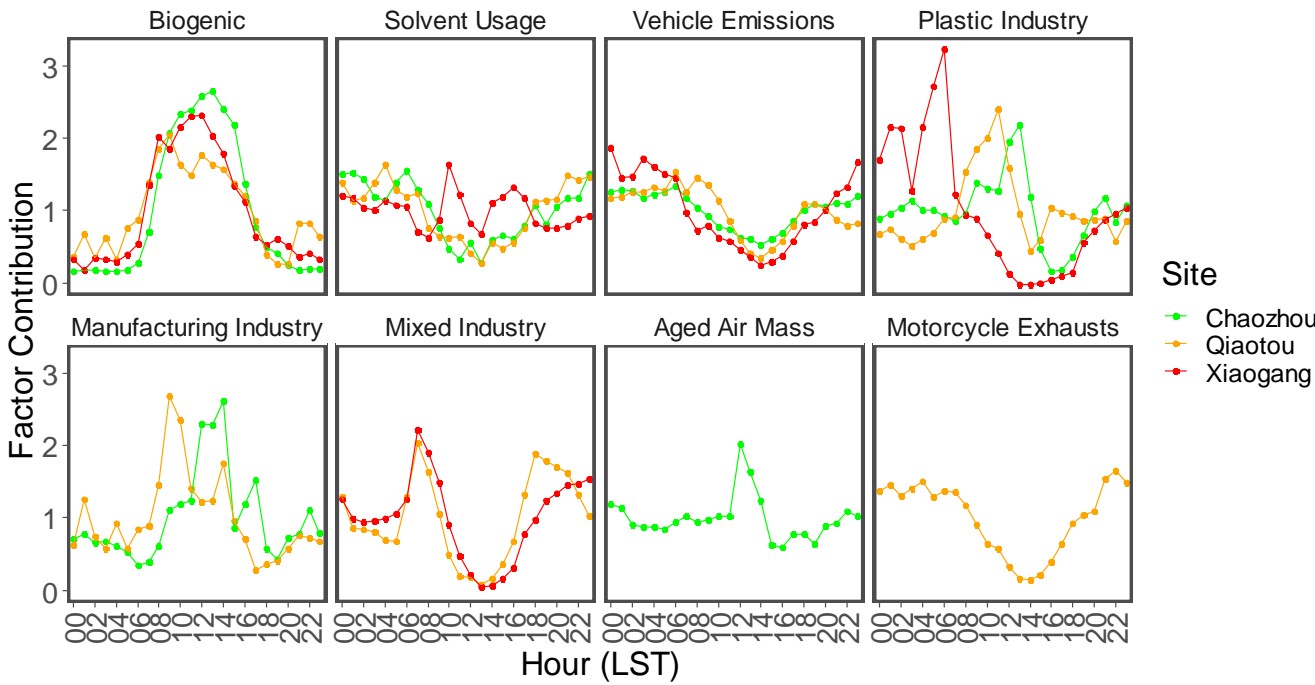

Figure 12: Diurnal variations of factor contribution calculated by the PMF model at Chaozhou, Qiaotou, Xiaogang PAMS stations




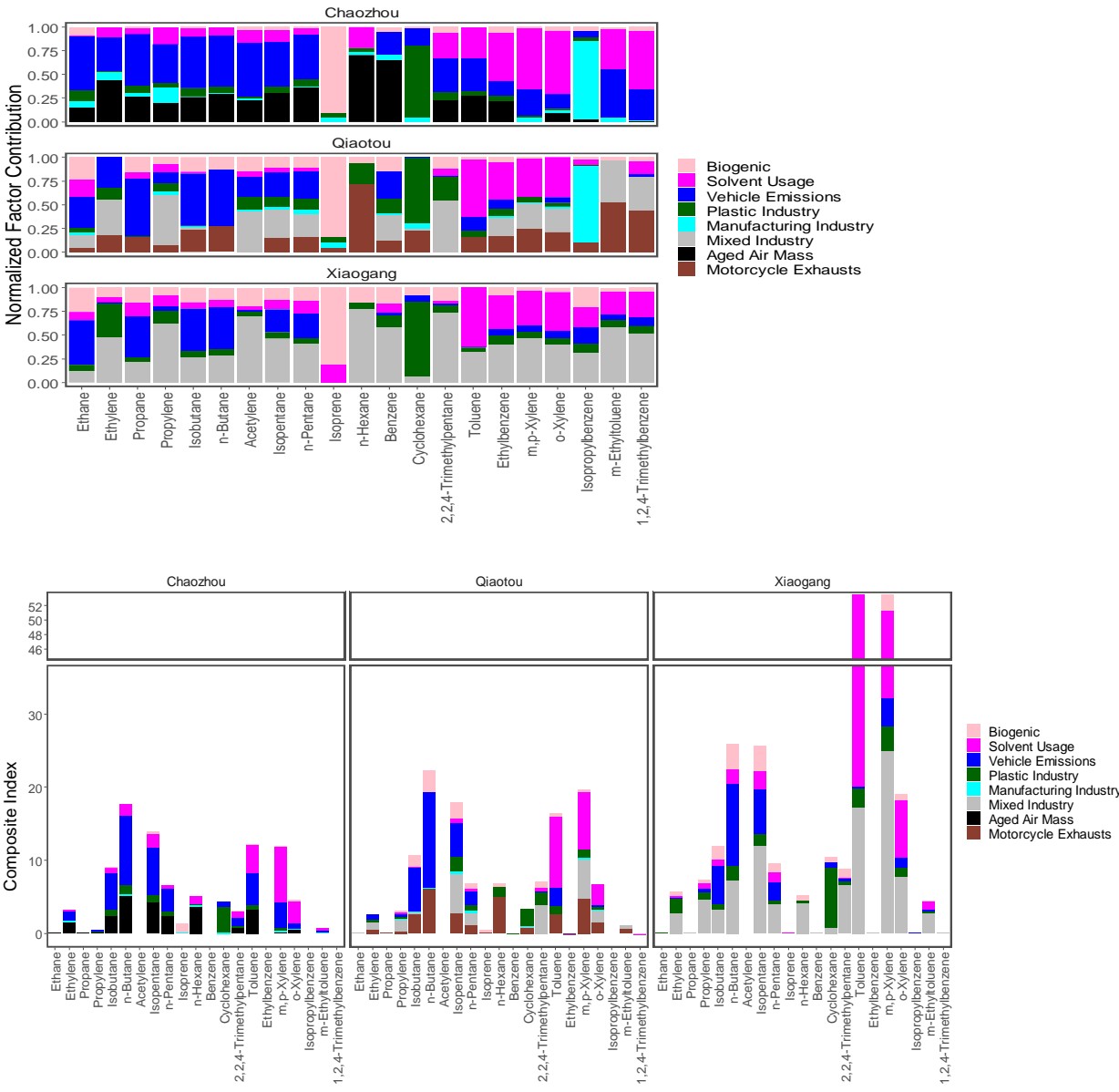

Figure 13: Normalized factor contribution and composite index of source profile to each PAMS-VOC species at Chaozhou, Qiaotou, Xiaogang PAMS station.

## 4. Conclusion

In this work, we used the CMAQ-DDM-3D sensitivity tool in conjunction with PMF model source apportionment to provide a comprehensive analysis of the major contributors to VOC species and the ozone formation potential (OFP) over an area of southern Taiwan. We developed a composite index that quantitatively combines the CMAQ-DDM sensitivity coefficient and PMF resolved factor contribution to identify the key VOC source-species for effective ozone abatement strategy, which is applicable to other urban areas that characterized by VOC-limited condition. A representative case in October 2018 with a daily 8-hour maxima $O_3 > 75$ ppb occurring often was the focus of this study. Low $NO_x$ levels and high BVOC levels in the inland areas yielded a $NO_x$-limited regime. Although reducing $NO_x$ emissions can reduce the inland $O_3$ concentrations, it could adversely increase urban $O_3$ due to the reduced titration effect. This is because the $O_3$ sensitivity production in urban area is mainly dominated by VOC-limited regime and further reducing $NO_x$ emissions can suppress the titration effect and eventually increased the urban $O_3$ concentration. In contrast, control of VOC emissions is effective in urban area to reduce $O_3$ concentrations and has no adverse effect in rural area. Our DDM sensitivity analysis identified the six most sensitive VOC species or groups: XYL (xylene), OLE (terminal olefins), PAR (paraffins), ETH (ethene), TOL (toluene), IOL (internal olefins) in descending order, which are mainly attributed to the petrochemical industry, painting & printing industry, and vehicle emissions. Based on a composite index, effective ozone abatement strategies should prioritize (1) solvent usage such as painting, coating and printing industry that emits abundant toluene and xylene, (2) petrol vehicular emissions with high compositions of n-butane, isopentane, isobutane and n-pentane, and (3) the petrochemical industry with emphasis on ethylene and propylene.

Besides, our DDM results also highlighted the important role of formaldehyde in OFP over the study area given that it is ranked the seventh highest VOC species for $O_3$ sensitivity and on a per-molecule basis for $O_3$ sensitivity is similar to alkene (i.e. OLE, ETH, IOL) and aromatics (XYL, TOL) species. Mapping of formaldehyde to the PMF source apportionment was not done due to the lack of oxygenates in the PAMS measurement inventories. Thus, oxygenates such as FORM, ALD2, ALDX, ACET, KET along with alcohols (ETOH and MEOH) should be included in the PMF source apportionment for more complete source identification in future studies. In the current work, WRF and CMAQ are resolved at high resolution 1 km

to best represent the features of local circulations (i.e land-sea breeze, urban heat island effect) at urban scale. However, simulation at 1 km is obviously too expensive for large domain or regional modelling. The differences in HDDM and PMF analysis between D03 (3 km) and D04 (1 km) due to grid resolution remain an open question which deserves future in-depth investigations. Although the performance of the simulated meteorological parameters (T2, WS, and WD) at both urban and

rural stations are acceptable in the benchmark recommended by USEPA, notable differences in temperature (underestimation) and wind speed (overestimation) are still observable in our simulation work. These biases could be susceptible to underestimation in photochemical ozone production due to the fictitious cold bias and enhanced dispersion. Therefore, careful treatment on the urban-scale data assimilation in temperature, wind field and relative humidity are recommended in future to improve the model prediction.


Other important findings obtained in this study are summarized below:

- Negative (positive) first-order sensitivities to daytime $NO_x$ emissions are dominant in urban (inland) area, indicating ozone production sensitivity favors the VOC-limited regime ($NO_x$-limited regime). Negative sensitivities are also extended to some parts of the coastal area of Pingtung county, reflecting the downwind transport of the urban $NO_x$

by the steering northeasterly winds due to the terrain effects and local circulations.

- Most of the urban areas (except Xiaogang, an industrial park) exhibited negative first-order and second-order sensitivities to daytime $NO_x$ emissions, indicating a negative $O_3$ convex response. However, due to the large negative second-order sensitivities, the urban $O_3$ increases from the linear effect are largely attenuated by the non-linear effect. As a result, urban $O_3$ becomes less sensitive to changes in $NO_x$ emissions but more sensitive to VOC

emissions which favors the VOC-limited conditions.

- Based on the PMF model source apportionment, a total of 8 factor profiles are identified as mixed industry (21%), vehicle emissions (22%), solvent usage (17%), biogenic (12%), plastic industry (10%), aged air mass (7%), motorcycle exhausts (7%), and manufacturing industry (5%).

- Benefit of VOC control in inland areas is expected to reduce gradually when $NO_x$ emissions continue to decrease over the long-term but control of VOCs in highly polluted urban areas remains effective despite a large reduction in $NO_x$ in the future.

**Code/Data Availability**

The code of the WRF software was obtained from https://www2.mmm.ucar.edu/wrf/users/download/. The code of the CMAQ software was taken from https://github.com/USEPA/CMAQ/. The FNL data were adopted from https://rda.ucar.edu/datasets/. The Positive Matrix Factorization Model for Environmental Data Analyses was obtained from https://www.epa.gov/air-research/positive-matrix-factorization-model-environmental-data-analyses. The observational data of surface meteorology, air pollutant and photochemical assessment monitoring stations (PAMS) were provided by Taiwan Environmental Protection Agency (TEPA) at https://www.epa.gov.tw.

**Author Contribution**

*Jackson Hian-Wui Chang*: Conceptualization, Methodology, Investigation, Formal analysis, Validation, Software, Writing - Original Draft. *Stephen M. Griffith*: Visualization, Writing- Reviewing and Editing. *Steven Soon-Kai Kong*: Formal analysis, Software. *Ming-Tung Chuang*: Formal analysis, Software. *Neng-Huei Lin*: Visualization, Funding acquisition, Writing- Reviewing and Editing.

**Competing Interests**

The authors declare that they have no conflict of interest

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
