# Peer review of "Development of a CMAQ-PMF-based composite index for prescribing an effective ozone abatement strategy: A case study of sensitivity of surface ozone to precursor VOC species in southern Taiwan"

_Atmospheric Chemistry and Physics, 2022_

## Referee Comment (RC1)

**Comments on acp-2022-691**

**General comments**

This manuscript presented a studying using CMAQ model with HDDM and PMF methods together to identify the contributions of VOCs to O3 at south part of Taiwan. The authors presented their efforts to conduct model simulations at 1km grid resolution to quantify the impacts of each VOC species from 8 different emission sectors, and also defined their own index to indicate the importance of each factor. Air pollution is always an critical issue for densely populated areas with intensive industrial facilities, and it's important to find the most economic-effective control strategy. This study made a contribution by providing in-depth investigation of O3 response to emissions changes, which could be an applicable suggestion for air quality management policy makers. The manuscript is well organized with solid method and data, and the results are presented and discussed thoroughly. Therefore I would recommend it to be published with minor revisions. My comments are listed as below.

Major Comment#1: The objective of this study was to identify the key VOC species and emission sectors to ozone abatement as the manuscript mentioned. So for the simulation period over a specific pollution event, is it able to represent typical ozone pollution conditions over southern Taiwan? Sec2.1 suggested the synoptic weather in fall usually favors ozone pollution due to less precipitation and weak vertical dispersion, but rest of the manuscript were mostly focused on impacts of emission. Therefore it is necessary to clarify if the conclusions of this study were applicable for fall ozone management only, as the driving factors and chemistry may differ in other seasons too. In addition, a brief description of ozone seasonality at southern Taiwan would be helpful to demonstrate how severe it is in fall season.

Major Comment#2: There was no description regarding chlorine emission in the manuscript, which could be a very important factor that may change the conclusion of this study, considering a vast part of simulation domain was over ocean. CMAQ was able to handle chlorine-related chemistry since earlier versions but usually it was applied over inland areas without too much attention devoted to oceanic chlorine emission. But for this study the emission from ocean may play an important role in ozone pollution over coastal cities in southern Taiwan.

**Detailed comments:**

(1). Did domain D03 simulation also applied HDDM and PMF? It would be interesting to reveal the differences between D03 and D04 due to grid resolution. 1km simulation is obviously too expensive for large domain, thus it would be helpful to see how well the 3km simulation can capture key factors to ozone formation.

(2). line145: Using 2016 TEDS emission is acceptable for 2018 simulation as there shall be no significant difference, but why not simulating a pollution in 2016 instead?

(3). Fig.10: Diurnal patterns of factor contribution are very different between sectors, for example, biogenic and motorcycle exhausts. Please provide a brief discussion.

(4). Was nudging turned on for WRF? Please clarify it in the manuscript.

(5). "DDM 3D" on line 47, "GCE" and "RRTM" on line 173 need clarifications.

(6). line 114-115 needs references.

(7). line112 "Oct" should be "Oct.".

(8). line266: Please clarify how was the threshold 0.5 determined

(9). Please using consistent denotes for models. Line134 suggested using CMAQv5.2.1, while line219 indicated using v5.2.

(10). Was HDDM approach only used in the experiments S3~5 when considering higher order sensitivity, while DDM approach is used for the rest experiments and in conjunction with PMF analysis? If so, please clarify these details in the manuscript.

(11). The spatial contribution of the O3 sensitivity to each VOCs component is related to the spatial pattern of each VOCs emissions as discussed on 413-416. Please include the several plots to show the spatial pattern of VOC emissions in the supporting information.

---

## Author Response (AR1)

**Comments on acp-2022-691**

We thank the reviewers for their constructive comments, which we believe have improved our manuscript. To facilitate the review process, our replies are highlighted in blue color font with *blue italicized font* representing text excerpts and ***blue bold italicized font*** representing new additions to the text. Please see our replies to each comment below.

**Anonymous Referee #1**

General comments This manuscript presented a studying using CMAQ model with HDDM and PMF methods together to identify the contributions of VOCs to O3 at south part of Taiwan. The authors presented their efforts to conduct model simulations at 1km grid resolution to quantify the impacts of each VOC species from 8 different emission sectors, and also defined their own index to indicate the importance of each factor. Air pollution is always an critical issue for densely populated areas with intensive industrial facilities, and it's important to find the most economic-effective control strategy. This study made a contribution by providing indepth investigation of O3 response to emissions changes, which could be an applicable suggestion for air quality management policy makers. The manuscript is well organized with solid method and data, and the results are presented and discussed thoroughly. Therefore I would recommend it to be published with minor revisions. My comments are listed as below.

We thank the reviewer for recognizing an important finding of our work.

Major Comment#1: The objective of this study was to identify the key VOC species and emission sectors to ozone abatement as the manuscript mentioned. So for the simulation period over a specific pollution event, is it able to represent typical ozone pollution conditions over southern Taiwan? Sec2.1 suggested the synoptic weather in fall usually favors ozone pollution due to less precipitation and weak vertical dispersion, but rest of the manuscript were mostly focused on impacts of emission. Therefore it is necessary to clarify if the conclusions of this study were applicable for fall ozone management only, as the driving factors and chemistry may differ in other seasons too. In addition, a brief description of ozone seasonality at southern Taiwan would be helpful to demonstrate how severe it is in fall season.

Response Major Comment #1

The selected case can reasonably represent the typical ozone pollution episodes in southern Taiwan because the synoptic weather pattern of the event features a weak intrusion of Asian continental anticyclone system with no apparent influence from Pacific subtropical high-pressure system (see Figure S1). According to Hsu & Cheng (2019), this synoptic weather pattern is conducive to ozone pollution episodes, typically occurs during the seasonal transition period and has the highest occurrence in October. The ozone seasonality in southern Taiwan is also briefly described. We addressed and clarified accordingly in the revised manuscript on line 114-129.

Line 114-129: *"The selected case in October 2018 is the seasonal transition period when the summer season is in transition to the winter. The case can reasonably represent the typical ozone pollution conditions during seasonal transition period in Taiwan because the synoptic weather pattern of the event features a weak intrusion of Asian continental anticyclone system which slowly propagated eastward causing the prevailing wind in Taiwan dominated by weak northeasterly (NE) flows due to continental high-pressure peripheral circulation (see Figure S1). Hsu & Cheng (2019) identified six synoptic weather patterns in Taiwan and studied the characteristics of corresponding air pollutants in each pattern using 6 years (2013-2018) daily averaged wind fields and sea-level pressure observed at surface weather stations in Taiwan. Among the six patterns (C1-C6), C4 has the highest mean $O_3$ concentrations and occurs predominantly in October. It features a weak anticyclone over the Asian continent and the Pacific subtropical high-pressure system does not have an apparent influence in Taiwan, which is similar to our selected case in October 2018. Although the photochemistry is strong in summer season, the seasonal $O_3$ variation in Taiwan shows that the monthly $O_3$ concentration is relatively higher during the seasonal transition months (i.e. October) compared to other seasons (Hsu and Cheng, 2019; Chen et al., 2021; Cheng et al., 2015). This is because during the seasonal transition months, when the photochemical reaction is still strong compared to that of the winter months together with the reduced ventilation capability, the $O_3$ concentration can accumulate (Yen and Chen, 2000; Tsai et al., 2008). (See Figure S2 for ozone seasonality in Taiwan)"*

[Figure]

*Figure S1: Synoptic weather pattern retrieved from NCEP-FNL reanalysis data valid at 00 UTC from 07 October 2018 to 23 October 2018 showing 850 hPa winds in vector referenced at 20 m s⁻¹ and sea level pressure in color contoured from 980 to 1020 hPa by 2 hPa. Taiwan is highlighted with green color.*

[Figure]

*Figure S2. (a) Annual averages of O₃ and NOₓ over a 19-year period at the Taipei, Taichung, Tainan and Kaohsiung sites. (b) Same observed data sets as (a) but with O₃ and NOₓ presented as monthly variations. (c) Number of high O₃ days (daily maximum O₃ > 100 ppbv) and the average of the top fifth-percentile O₃ concentration in each year. (d) Similar to (c) but presenting the occurrence of days with O₃ > 100 ppbv as monthly variations. Unit for concentrations is in ppbv. Source: (Cheng et al., 2015)*

Major Comment#2: There was no description regarding chlorine emission in the manuscript, which could be a very important factor that may change the conclusion of this study, considering a vast part of simulation domain was over ocean. CMAQ was able to handle chlorine-related chemistry since earlier versions but usually it was applied over inland areas without too much attention devoted to oceanic chlorine emission. But for this study the emission from ocean may play an important role in ozone pollution over coastal cities in southern Taiwan.

Response Major Comment #2

Thank you for the comment. The oceanic chlorine emission in our simulation is handled by online CMAQ module which is calculated based on an OCEAN file, whereas the anthropogenic chlorine emission is obtained from TEDS v10 emission inventory. We clarified accordingly in the revised manuscript on line 174-182 and 214-216.

Line 174-182: *"In our work, the oceanic chlorine emission is calculated online by CMAQ as a function of meteorology using an OCEAN file which specifies the fraction of each grid cell that is open ocean (OPEN) or surf zone (SURF). Figure 2a-c presents the spatial distribution of CMAQ calculated sea-salt aerosol cations (ASEACAT - $Na^+$, $K^+$, $Ca^{2+}$ and $Mg^{2+}$), fine-mode chlorine and coarse-mode chlorine emission rates averaged during the entire simulation period. The sea-spray emissions were higher in the surf zone area and highest emission rates were found along the eastern offshore of southern Taiwan. This is because of the enhanced formation of sea-spray aerosols associated with higher relative humidity and greater offshore winds along the eastern offshore of southern Taiwan that is open to the Western Pacific Ocean. Besides, the anthropogenic chlorine emissions (PCL) are obtained from TEDS v10 emissions, and they are concentrated over the heavily industrialized urban areas of southern Taiwan (see Figure 2d)."*

Line 214-216: *"The halogen chemistry in CB6 considers chlorine-related reactions such as $ClNO_2$, $HCl$ and $HNO_3$ production from heterogeneous uptake of $N_2O_5$ on the aerosol surface, which are important to ozone pollution over coastal cities in southern Taiwan."*

[Figure]

*Figure 2: CMAQ calculated (a) sea-salt aerosol cations (ASEACAT) emissions (Na⁺, K⁺, Ca²⁺ and Mg²⁺), (b) fine-mode chlorine SSA emissions, (c) coarse-mode chlorine SSA emissions and (d) TEDS v10 anthropogenic chlorine (PCL) emissions averaged during the entire simulation period.*

Detailed comments:

(1). Did domain D03 simulation also applied HDDM and PMF? It would be interesting to reveal the differences between D03 and D04 due to grid resolution. 1km simulation is obviously too expensive for large domain, thus it would be helpful to see how well the 3km simulation can capture key factors to ozone formation.

Response Detailed Comment #1

Thank you for the comment. We acknowledge that the differences between D03 and D04 are interesting to reveal, unfortunately HDDM is turned off for domain D01-D03 to save the expensive computation cost. Besides, we also think that this issue deserves a standalone topic to be explored in the future. Therefore, we carefully address this issue as future work in the conclusion.

Line 698-703: *"In the current work, WRF and CMAQ are resolved at high resolution 1 km to best represent the features of local circulations (i.e land-sea breeze, urban heat island effect) at urban scale. However, simulation at 1 km is obviously too expensive for large domain or regional modelling. The differences in HDDM and PMF analysis between D03 (3 km) and D04 (1 km) due to grid resolution remain an open question which deserves future in-depth investigations."*

(2). line145: Using 2016 TEDS emission is acceptable for 2018 simulation as there shall be no significant difference, but why not simulating a pollution in 2016 instead?

Response Detailed Comment #2

We selected the case 2018 for 2016 TEDS v10 emission because TEDS emission inventory is updated every 3 years, in which the next update is 2019 TEDS v11 emission. At the time of preparing the work, 2016 TEDS v10 emission is the latest emission inventory readily available in gridded format for CMAQ simulation. Therefore, we selected the case 2018 to represent the latest year useable for 2016 TEDS v10 emission. In addition, 2018 case is also more representative of Taiwan local's emissions with less influence from LRT of China's emissions.

(3). Fig.10: Diurnal patterns of factor contribution are very different between sectors, for example, biogenic and motorcycle exhausts. Please provide a brief discussion.

Response Detailed Comment #3

Thank you for the comment. Please see line 580-588 for the additional brief discussion.

Line 580-588: *"The diurnal pattern of factor distribution of biogenic sector displays a clear diurnal cycle peak at the noontime 10-12 LST in all three stations. Meanwhile, the motor exhaust and vehicle emissions sector peak during the morning and evening traffic rush hours. The aged airmass sector is only identified in CZ and its hourly factor contribution mostly existed at more stable levels when compared to other factors except for an obvious peak at 12 LST due to the sea-breeze penetration that pushes the urban polluted aged air mass towards inland area. The diurnal pattern related to industrial activity exhibits different peak hours depending on sector and station. For instance, the hourly factor contribution of solvent usage in XG has a clear bimodal peak at 10 and 16 LST; plastic industry in QT and CZ has a clear diurnal cycle peak in the noon time at 11 and 13 LST; mixed industry in QT and XG has a clear bimodal peak at 07 and 18 LST."*

(4). Was nudging turned on for WRF? Please clarify it in the manuscript.

Response Detailed Comment #4

Yes, grid nudging was turned on for WRF in the coarse domain D01 and D02 while observation nudging was turned on in the fine domain D03 and D04. Details of the nudging technique are clarified in the revised manuscript on line 198-207. Please see below revision.

Line 198-207: *"To obtain more accurate dynamical downscaling, grid nudging was applied in the coarse domain D01 and D02; observation nudging was applied in the fine domain D03 and D04. Grid nudging is applied to the horizontal wind components, potential temperature, and water vapor mixing ratio; it is only applied above the PBL. The observational data for observation nudging include hourly surface observations such as atmospheric pressure,*

*air temperature, relative humidity, wind speed and wind direction from 36 surface meteorological stations (https://www.epa.gov.tw/), and the twice-daily at 00:00 and 12:00 UTC sounding data such as potential height, temperature, dew point temperature, RH, wind direction, wind speed at each specified isobaric level from 2 radiosonde observation stations in Taiwan. The nudging coefficients, which determine the strength of the assimilation tendency term were set to be $6 \times 10^{-4}$ for observation nudging and $3 \times 10^{-4}$ for grid nudging. These values of coefficients were recommended by the WRF user guide and tested to be appropriate in previous studies (Li et al., 2022; Borge et al., 2008)."*

(5). "DDM 3D" on line 47, "GCE" and "RRTM" on line 173 need clarifications.

Response Detailed Comment #5

Revised.

Line 47: "…***Decoupled Direct Method in Three Dimensions*** *(DDM-3D)…*"

Line 227-228: *"Also used were the **Goddard Cumulus Ensemble (GCE)** microphysics scheme (Tao et al., 2003), **Rapid Radiative Transfer Model (RRTM)** longwave radiation scheme (Gallus and Bresch, 2006) …"*

(6). line 114-115 needs references.

Response Detailed Comment #6

Revised.

Line 130: *"During the seasonal transition period in autumn, southern Taiwan often suffers from high O3 episodes (**Hsu and Cheng, 2019; Chen et al., 2004, 2021**)."*

(7). line112 "Oct" should be "Oct.".

Response Detailed Comment #7

Revised and all other occurrences are also revised accordingly.

Line 113: *"In this study, we selected the period from 07-20 **Oct.** 2018…"*

Line 137: *"A 5-day spin-up period (02-06 **Oct.** 2018) was discarded…"*

(8). line266: Please clarify how was the threshold 0.5 determined

Response Detailed Comment #8

This threshold value 0.5 was determined according to the EPA PMF v5.0 user guide. Please see the clarification below.

Line 330-332: "***This threshold value 0.5 was determined according to the EPA PMF v5.0 user guide and also recommended by other PMF studies (Rajput et al., 2016; Reff et al., 2007)".***

(9). Please using consistent denotes for models. Line134 suggested using CMAQv5.2.1, while line219 indicated using v5.2.

Response Detailed Comment #9

Revised.

Line 287: *"In this study, CMAQ-HDDM-3D **v5.2.1** is used to…"*

(10). Was HDDM approach only used in the experiments S3~5 when considering higher order sensitivity, while DDM approach is used for the rest experiments and in conjunction with PMF analysis? If so, please clarify these details in the manuscript.

Response Detailed Comment #10

Yes, we clarified accordingly in the revised manuscript on line 294-296.

Line 294-296: *"Noted that HDDM approach was only used in experiment S3-S5 which involves calculation of higher order sensitivity, while DDM approach was used in all other experiments and in conjunction with PMF analysis."*

(11). The spatial contribution of the O3 sensitivity to each VOCs component is related to the spatial pattern of each VOCs emissions as discussed on 413-416. Please include the several plots to show the spatial pattern of VOC emissions in the supporting information

Response Detailed Comment #11

Thank you for the comment. We agree that the spatial distribution of $O_3$ sensitivity is related to the spatial distribution of each VOCs emissions. We included several plots of the spatial distribution of some highly sensitive VOC component emissions (i.e. XYL, OLE, PAR, ETH, TOL, IOL) in the supplementary material.

Line 483-486: **"To render a clearer picture on the spatial distribution of $O_3$ sensitivity to each VOCs component, the spatial distribution of some highly sensitive VOC component emissions (i.e. XYL, OLE, PAR, ETH, TOL, IOL) are provided in the supplementary material (see Figure S7)."**

[Figure]

*Figure S7: Spatial distribution of some highly sensitive VOC component emissions (i.e. XYL, OLE, PAR, ETH, TOL, IOL) at 12 LST.*

**Anonymous Referee #2**

The manuscript utilizes CMAQ model with HDDM and PMF methods to study the contributions of VOCs and O3 in Southern Taiwan. The High-resolution HDDM with indepth analysis of VOCs and NOx relationship on ozone formation has been presented. The article is well-written. I would recommend it with minor revisions. See my comments below.

We thank the reviewer for recognizing an important finding of our work.

Comments#1 Some discussion on the contribution of long-range transport or from a non-Taiwan background contribution should be included to allow readers to better understand the air quality situation in Taiwan. As the selected case is in October, a large influence may be exhibited from non-Taiwan sources (e.g., Chinese emissions). Moreover, the recent reduction in China may contribute more to Taiwan than the local reduction. Therefore, it is worth including some discussion.

Response Comment #1

Thank you for the comment. We acknowledge the substantial contribution of long-range transport to air quality in northern and central Taiwan especially under strong northeasterly wind conditions. However, the contribution of long-range transport is relatively smaller in southern Taiwan due to the geographic blocking. For clarity, we included a brief discussion pertaining to this issue in the revised manuscript.

*Line 164-172: "Other than local anthropogenic emissions, the contribution of long-range transport (LRT) from East Asia (e.g. Chinese emissions) is also substantial to Taiwan's air quality especially under strong northeasterly winds condition (Wu and Huang, 2021; Chang et al., 2022a). Lin et al. (2004) identified three types of common LRT events in Taiwan: (1) dust storm (DS), LRT with pollutants (frontal pollution; FP), and (3) LRT of background airmasses (BG). When there is no frontal system, local pollution (LP) dominates the air quality in Taiwan. During wintertime and springtime, the occurrence of LP cases were 70% and about 30% were LRT cases (Lin et al., 2004). Lin et al. (2005) estimated that the long-range transport of pollutants contributes to about 30 μg m⁻³, 230 ppb and*

*0.5 ppb to the PM₁₀, CO, and SO₂ concentrations, respectively, in northern and eastern Taiwan. Meanwhile a*

*smaller contribution is estimated in southern Taiwan due to the geographic (Lin et al., 2005)".*

Comments#2 The definition of "urban" and "inland rural area" was not clearly defined. The location or boundary was not clear. The general meteorological pattern (i.e., wind direction) for the event and topology (terrain location) were unclear. Although it has been presented in tables, it is hard for readers to understand the general transport pattern between urban and inland rural areas (e.g., the Influence of sea breeze, or terrain on wind speed and direction). They are important transport mechanisms influencing O3 formation locally. The authors used abbreviations (e.g., XG, CZ etc.), but it wasn't labelled in the figures. It makes it difficult to understand. It is suggested to mark it directly in the figures (e.g., Fig 3, 4 and 6, *S4, S10*).

Response Comment #2

We apologized for the confusion on the definition of "urban" and "inland" area, which was not clearly defined in the previous manuscript. We clarified accordingly in Section 2.2 WRF-CMAQ Model Configuration. For the general meteorological pattern of the event, we addressed accordingly in an additional paragraph in Section 2.1 Study Period & Area on line 238-248. The paragraph also briefly describes the transport mechanism of urban O₃ by local circulations (i.e. land-sea breeze, wind speed and wind direction) with an additional Figure 3. The topography of the innermost domain is also shown in additional Figure 2. After revision, the abbreviations (e.g., XG, CZ, and QT) are now labelled in Fig 5, 6 and 8, S6, S13. Please see the revised figures below.

Line 238-248: *"The "urban" and "inland" grid cells are defined according to the USGS-24 Land Use Category. "Urban" area is represented by Class 1 - Urban and Built-up Land, which we further classified into Class 31, 32, 33 (see Figure 1b) for WRF Single-Layer Urban Canopy Scheme (SLUCM) simulation. We refer the readers to our previous work for detailed discussion on the land use classification and SLUCM implementation in Chang et al. (2022b). "Inland" area is represented by Class 6 – Cropland/Woodland Mosaic (see Figure 1b). The general meteorological pattern of the event features a weak intrusion of Asian continental anticyclone system which slowly propagated eastward causing the prevailing wind at synoptic scale in Taiwan dominated by weak northeasterly (NE) flows due to continental high-pressure peripheral circulation (see Figure S1). At local scale in southern*

*Taiwan, the steering of weak NE flows by the orographic effect of the Central Mountain Range (CMR) enhanced the local circulations (i.e. land-sea breeze), and eventually pushed the locally produced urban O₃ as well as its precursors NO$_x$ and NMHC towards the inland areas (see Figure 3)."*

[Figure]

*Figure 1: (a) Domain configuration of four-nested grid system, (b) land use of the innermost domain;* **"urban" and "inland" areas are represented by Class 31, 32, 33 and Class 6, respectively**, *(c,d) monthly averaged NO$_x$ and VOC emissions in the innermost domain obtained from 2016 TEDS-10 emission inventory. The location of each TEPA air quality stations (red stars) and PAMS stations (red dots with label) used in the current study are displayed in the innermost domain. Refer Figure S1 and Table S3 for details of each TEPA and PAMS station.*

[Figure]

*Figure S1: Synoptic weather pattern retrieved from NCEP-FNL reanalysis data valid at 00 UTC from 07 October 2018 to 23 October 2018 showing 850 hPa winds in vector referenced at 20 m s$^{-1}$ and sea level pressure in color contoured from 980 to 1020 hPa by 2 hPa. Taiwan is highlighted with green color.*

[Figure]

*Figure 3: (a) Topography of the innermost domain. (b, c) Spatial distribution of O₃ concentration averaged during the entire simulation at daytime 10 LST and nighttime 20 LST, respectively. (d-f) Vertical profile of NMHC concentration cross sectioned at AB (see Figure 2a) averaged during the entire simulation at 10 LST, 15 LST, and 20 LST, respectively. (h-i) Same as d-f but for NO₂ concentration.*

[Figure]

*Figure 5: CMAQ-HDDM first-order sensitivity coefficient of $O_3$ to (a) $NO_x$ emissions, (b) VOC emissions, second-order sensitivity coefficient of $O_3$ to (c) $NO_x$ emissions, (d) VOC emissions, (e) second-order cross sensitivity coefficients of $O_3$ to $NO_x$, VOC emissions, at daytime 09-15 LST averaged during the entire simulation period. Magenta and green highlighted borderline represents Xiaogang District and Pingtung region, respectively.*

[Figure]

*Figure 6: Spatial distribution of O₃ concentration in (a) baseline with no perturbations in NOₓ and VOC emissions, and changes in O₃ concentration under (b) NOₓ control scenario, (c) VOC control scenario, and (d) NOₓ & VOC control scenario at daytime 12 LST. All scenarios reduced the targeted emissions by 5% except for the baseline.*

[Figure]

*Figure 8: (a) Daily averaged CMAQ-DDM first-order sensitivity coefficient of O₃ concentrations calculated per number of grids to each modelled VOC species arranged in ascending order for urban and inland area. Sensitivity of O₃ and ozone formation potential (OFP) to (b,e) alkenes emissions (OLE + ETH + IOL), (c,f) aromatics emissions (XYL + TOL), and (d,g) alkanes emissions (PAR) at 12 LST.*

[Figure]

*Figure S6: Spatial distribution of ratio NO$_x$ / VOC averaged at 12:00 LST during the entire simulation period.*

[Figure]

*Figure S13: Spatial distribution of (a) daily 8h maxima O₃, (b) occurrence of daily 8h maxima O₃ >75 ppb (c) daily maxima NO₂, (d) daily maxima VOC, averaged during the entire simulation period at the lowest model level in the innermost domain.*

Comments#3 What will be the impact on the O3 and VOCs analysis under high underestimation in temperature and overestimation in wind speed in urban shown in Fig S8? It is recommended to have more discussion on the base-case model performance. Will the overestimated NO2 create VOC limited situation in Urban (Fig S9), instead of emissions? How much error will it create? What will be the impact of overestimated NO2 on HDDM and PMF analysis?

Response Comment #3

Thank you for the comment. The discussion on the base-case model performance is provided in supplementary material (see Supplementary Material – Model Evaluations). We addressed the possible impact on the $O_3$ and VOC analysis under notable differences in temperature and wind speed on line 702-707. For the overestimated $NO_2$ in urban areas, we believe that such small error (MB = 5.5 ppb, MNB = 28%) should have minimal impact on HDDM and PMF analysis and is unlikely to create VOC-limited condition, instead of emissions. We showed in the Taylor-series approximation analysis when we hypothetically reduce $NO_x$ emission at arbitrary -50% and -25% scenario, which should expect an improvement in the overestimation of $NO_2$, urban $O_3$ at XG remains in a VOC-limited condition. Therefore, it is certain that the VOC-limited condition at urban areas of southern Taiwan is mainly due to the high anthropogenic $NO_x$ emissions. We addressed this issue accordingly in the revised manuscript on line 439-442.

Line 702-707*: "Although the performance of the simulated meteorological parameters (T2, WS, and WD) at both urban and rural stations are acceptable in the benchmark recommended by USEPA, notable differences in temperature (underestimation) and wind speed (overestimation) are still observable in our simulation work. These biases could be susceptible to underestimation in photochemical ozone production due to the fictitious cold bias and enhanced dispersion. Therefore, careful treatment on the urban-scale data assimilation in temperature, wind field and relative humidity are recommended in future to improve the model prediction."*

Line 439-442*: "We showed in the Taylor-series approximation analysis when we hypothetically reduce $NO_x$ emission at arbitrary -50% and -25% scenario, which should expect an improvement in the overestimation of $NO_2$, urban $O_3$ at XG remains in a VOC-limited condition. Therefore, it is certain that the VOC-limited condition at urban areas of southern Taiwan is mainly due to the high anthropogenic $NO_x$ emissions."*

Comments#4 There are many colour labels in Fig S1. However, they were not defined. Please also add (QT, CZ, and XG – abbreviation) into Fig S1.

Response Comment #4

After revision, the color labels are defined and the abbreviations (QT, CZ, XG) are added in Figure S3.

[revised manuscript text omitted]